# Iron Redox Cycling in Persulfate Activation: Strategic Enhancements, Mechanistic Insights, and Environmental Applications—A Review

**DOI:** 10.3390/nano15221712

**Published:** 2025-11-12

**Authors:** Zutao Zhang, Fengyang Du, Hongliang Shi, Huanzheng Du, Peiyuan Xiao

**Affiliations:** 1College of Environmental Science and Engineering, Tongji University, Shanghai 200092, China; xydst0@gmail.com; 2Circular Economy Research Institute, School of Marxism, Tongji University, Shanghai 200092, China; 3Yangtze River Delta Institute of Circular Economy Technology, Jiaxing 314001, China; alyssadufy@gmail.com (F.D.); 15382356086@163.com (H.S.); 4State Key Laboratory of Pollution Control and Resources Reuse, College of Environmental Science and Engineering, Tongji University, Shanghai 200092, China

**Keywords:** iron-based catalysts, persulfate activation, iron cycling, electron transfer, environmental remediation, reaction mechanisms

## Abstract

Iron-based catalysts for peroxymonosulfate (PMS) and peroxydisulfate (PDS) activation represent a cornerstone of advanced oxidation processes (AOPs) in environmental remediation, prized for their cost-effectiveness, environmental compatibility, and high catalytic potential. These catalysts, including zero-valent iron, iron oxides, and iron-organic frameworks, activate PMS/PDS through heterogeneous and homogeneous pathways to generate reactive species such as sulfate radicals (SO_4_•^−^) and hydroxyl radicals (•OH). However, their large-scale implementation is constrained by inefficient iron cycling, characterized by sluggish Fe^3+^/Fe^2+^ conversion and significant iron precipitation, leading to catalyst passivation and oxidant wastage. This comprehensive review systematically dissects innovative strategies to augment iron cycling efficiency, encompassing advanced material design through elemental doping, heterostructure construction, and defect engineering; system optimization via reductant incorporation, bimetallic synergy, and pH modulation; and external field assistance using light, electricity, or ultrasound. We present a mechanistic deep-dive into these approaches, emphasizing facilitated electron transfer, suppression of iron precipitation, and precise regulation of radical versus non-radical pathways. The performance in degrading persistent organic pollutants—including antibiotics, per- and polyfluoroalkyl substances (PFASs), and pesticides—in complex environmental matrices is critically evaluated. We further discuss practical challenges related to scalability, long-term stability, and secondary environmental risks. Finally, forward-looking directions are proposed, focusing on rational catalyst design, integration of sustainable processes, and scalable implementation, thereby providing a foundational framework for developing next-generation iron-persulfate catalytic systems.

## 1. Introduction

The rapid advancement of industrialization and urbanization has led to the widespread contamination of water and soil environments by refractory organic pollutants, including antibiotics, dyes, pesticides, and PFAS [1,2,3]. These contaminants pose severe risks to ecosystems and human health due to their high toxicity, chemical stability, and low biodegradability. Conventional treatment techniques—such as biodegradation, adsorption, and physical separation—often prove ineffective for complete removal and may entail high operational costs or secondary pollution concerns. Consequently, there is an urgent need to develop efficient, economical, and environmentally sustainable degradation technologies.

Advanced oxidation processes (AOPs) have emerged as a promising strategy for deep mineralization of organic pollutants through the generation of reactive oxygen species (ROS). However, conventional Fenton reactions suffer from inherent limitations, including a narrow operational pH range (typically 2–4), substantial iron sludge production, and risks associated with the storage and transportation of H_2_O_2_ [4,5,6]. These drawbacks have motivated the development of novel AOPs that exhibit high efficiency, broad pH applicability, and minimal environmental impact.

Persulfate-based AOPs, utilizing peroxymonosulfate (PMS) or peroxydisulfate (PDS), have garnered considerable interest as viable alternatives. Persulfates offer advantages such as high stability, facilitating safer handling and transport, as well as high redox potentials that enable effective degradation of various recalcitrant organics [1,3]. Upon activation, persulfates generate multiple reactive species, primarily SO_4_•^−^ and •OH, which exhibit strong oxidative capacity and broad substrate applicability. Despite their potential, persulfates alone are incapable of degrading pollutants; they require catalytic activation to cleave the peroxy bond and produce ROS. Activation methods include transition metal catalysis, carbon-based materials, and energy-based processes such as UV and heating [7,8,9,10].

Among these, iron-based catalysts—such as zero-valent iron (ZVI), iron oxides, and iron-organic frameworks—are regarded as ideal activators due to the abundance, low cost, and low toxicity of iron [11,12,13,14,15]. Nevertheless, their catalytic efficiency is severely hampered by low iron cycling efficiency [11,12,13,14,15]. While Fe^2+^ rapidly activates PMS/PDS to yield SO_4_^•−^/•OH, it is subsequently oxidized to Fe^3+^ (with rate constants k in the range of 10–100 M^−1^s^−1^). The reduction of Fe^3+^ back to Fe^2+^ is notably slow (k < 0.001–1 M^−1^s^−1^), particularly in heterogeneous systems [10,13,16,17,18,19]. Moreover, under neutral or alkaline conditions, Fe^3+^ tends to form iron (oxyhydr)oxide precipitates (e.g., FeOOH, Fe(OH)_3_), which encapsulate active sites and impede the catalytic cycle. These issues result in inefficient oxidant utilization, incomplete pollutant mineralization, and limited pH applicability (often requiring acidic conditions), significantly hindering practical implementation.

There has been a surge of research interest in persulfate activation using iron-based catalysts. Bibliometric analysis of the Web of Science Core Collection reveals a marked upward trend in annual publications focused on iron-persulfate systems, underscoring the growing importance of this technology (Figure 1a). Keyword co-occurrence network analysis further indicates that enhancing iron cycling efficiency is a core research direction in iron-mediated persulfate-based AOPs to achieve the improvement of process performance and efficient removal of organic pollutants (Figure 1b). This growing research interest has yielded significant advances in understanding and enhancing the iron redox cycle. Recent studies have demonstrated that atomic-level engineering through single-atom Fe-N_4_ configurations can achieve exceptional persulfate activation while minimizing metal leaching [20]. Concurrently, heterointerface construction, such as in MoS_2_/Fe_3_O_4_ composites, has proven effective in facilitating interfacial electron transfer and accelerating Fe^3+^/Fe^2+^ conversion [21]. Defect engineering through the creation of oxygen or sulfur vacancies has emerged as another powerful strategy to modulate local electron density and optimize persulfate adsorption [22,23]. Furthermore, sustainable approaches utilizing waste-derived materials, such as catalysts synthesized from spent lithium-ion batteries, have shown promising performance while addressing environmental sustainability concerns [24,25]. These developments represent the cutting edge of iron redox cycling research and provide deeper mechanistic insights into the interfacial processes governing persulfate activation.

Despite the existence of previous reviews on iron-based persulfate activation, this work distinguishes itself by providing a unified and critical analysis centered explicitly on overcoming the iron cycling bottleneck. Unlike prior works that often catalog materials or applications, this review establishes a comprehensive “problem identification-strategy development-mechanism elucidation-application evaluation” framework. Our unique synthesis integrates enhancement strategies across catalyst design, system optimization, and external field assistance into a single mechanistic narrative, revealing how disparate approaches ultimately converge on accelerating electron transfer and suppressing iron deactivation. Furthermore, we place a distinct emphasis on establishing structure-activity relationships and on the practical implications of these mechanisms in complex environmental matrices, topics often treated superficially. This review aims to provide not only a summary of the state-of-the-art but also a foundational guide for the rational design of next-generation iron-persulfate systems.

## 2. The Root Causes and Impacts of Low Iron Cycling Efficiency

### 2.1. Thermodynamic and Kinetic Limitations

The inefficiency of the Fe^2+^/Fe^3+^ redox cycle, central to persulfate activation, arises from deeply intertwined thermodynamic and kinetic constraints (Figure 2). Thermodynamically, the reduction of Fe^3+^ to Fe^2+^ must overcome a substantial energy barrier, as reflected in its high standard reduction potential (E°(Fe^3+^/Fe^2+^) = +0.77 V vs. NHE). In persulfate-driven systems, the frequent absence of efficient endogenous electron donors promotes the hydrolysis of Fe^3+^ into insoluble and thermodynamically stable precipitates, such as Fe(OH)_3_ or FeOOH, which exhibit exceedingly low solubility constants (Ksp ≈ 10^−39^). This precipitation effectively depletes the dissolved Fe^2+^ pool, trapping the iron cycle in a kinetically inert, oxidized state. Furthermore, although high-valent iron species (e.g., Fe(IV)=O) possess high oxidative potential, their generation requires overcoming considerable activation energy barriers and often proves unsustainable without appropriate ligand stabilization or confined coordination environments [26,27,28]. Essentially, the system becomes thermodynamically “locked” in the Fe^3+^ state, stalling persistently in the absence of an efficient electron-compensation mechanism.

Kinetically, the iron cycle is impeded by three major bottlenecks. First, in homogeneous systems, the apparent rate constant (k) for the reduction of Fe^3+^ to Fe^2+^ decreases by 2–3 orders of magnitude as pH increases beyond 5, largely due to the predominance of less reactive hydrolyzed species such as Fe(OH)^2+^ [12]. Second, in heterogeneous catalytic systems, the accumulation of passivating layers (e.g., FeOOH) leads to an exponential increase in charge transfer resistance (often exceeding 10-fold after multiple cycles), severely impeding electron transport from the bulk material to reactive surface sites [29,30]. Third, radical scavenging introduces a self-consuming pathway: sulfate (SO_4_^•−^) and hydroxyl (•OH) radicals react with Fe^2+^ at near diffusion-limited rates (k ≈ 10^9^ M^−1^ s^−1^), significantly outpacing their reactions with target pollutants [31]. This leads to rapid consumption of Fe^2+^ on a microsecond timescale, macroscopically manifesting as a “kinetic short-circuit” within the iron cycle. Together, these kinetic barriers create a “rate trap” that drastically diminishes catalytic efficiency, even under otherwise thermodynamically favorable conditions.

The inhibition of iron cycling triggers a rapid decline in the system oxidation-reduction potential (ORP), yielding several critical consequences: (1) A drastic reduction in oxidant utilization: Depletion of Fe^2+^ causes persulfate activation rates to drop by 80–90%, resulting in an exponential decay in SO_4_•^−^ yield and extending the half-lives of pollutant degradation by 1–2 orders of magnitude [32,33,34]. (2) A shift in reaction pathways: The proportion of high-valent iron species (e.g., Fe(IV)=O) and non-radical oxidation pathways increases. Although capable of degrading certain contaminants, these pathways often exhibit poor selectivity toward chlorinated and nitrogenous organics, frequently yielding more toxic intermediates (e.g., chlorinated quinones and nitro-byproducts) [27,35,36]. (3) A cascade of ecotoxicological effects: Fe(OH)_3_ flocs can adsorb toxic transformation products, forming “iron-toxin” co-precipitates that accumulate in sediments and pose a secondary pollution risk. Moreover, these flocs may catalyze the formation of persistent free radicals (PFRs), initiating secondary chain reactions that further amplify environmental toxicity [37,38,39,40]. In summary, inefficient iron cycling not only compromises treatment performance but also exacerbates ecological risks, creating a detrimental negative feedback loop between reduced efficiency and heightened toxicity.

### 2.2. Challenges of Fe^3+^ Precipitation

Iron precipitation is a major contributor to the inefficient iron cycling observed in iron-based advanced oxidation processes. Common precipitates include iron (oxyhydr)oxides (e.g., FeOOH, Fe(OH)_3_), mixed-valence oxides (e.g., Fe_3_O_4_, γ-Fe_2_O_3_), and co-precipitates with anions such as phosphates or carbonates. These precipitates impede the iron cycle through multiple concurrent mechanisms. Primarily, precipitate formation drastically reduces the accessibility of active iron sites. For instance, a goethite (α-FeOOH) shell thicker than 3 nm can completely block Fe^2+^ diffusion, increasing charge transfer resistance (Rₜ) by 1–2 orders of magnitude and severely diminishing redox cycling efficiency [41,42]. Secondly, precipitation alters local iron redox thermodynamics. The formation of FePO_4_, for example, depresses the standard Fe^3+^/Fe^2+^ redox potential from +0.77 V to +0.36 V, thermodynamically suppressing Fe^2+^ regeneration [39]. Additionally, surface-adsorbed Fe(II) species are rapidly oxidized by sulfate radicals (SO_4_^•−^; k ≈ 4.6 × 10^9^ M^−1^s^−1^), disrupting the balance between surface and solution iron cycles and macroscopically reducing oxidant utilization by 70–80% [15,43].

The formation of iron precipitates is influenced by several environmental and operational factors, including pH, anion concentration, and catalytic cycling number. For example, as pH increases from 5 to 8, the formation rate of FeOOH rises significantly, with a pseudo-first-order precipitation rate constant (k) reaching 0.03 min^−1^ [15]. High concentrations of H_2_PO_4_^−^ and HCO_3_^−^ promote the formation of FePO_4_ and FeCO_3_ co-precipitates, markedly decreasing Fe(II) availability and further hindering the catalytic cycle [39]. With repeated catalytic cycles, the accumulation of precipitates on the catalyst surface leads to progressive deactivation, posing a significant challenge for long-term operation.

In summary, iron precipitation severely impedes the iron cycle through two interconnected mechanisms: (1) “thermodynamic locking”, where the formation of stable, insoluble iron (oxyhydr)oxide precipitates (e.g., FeOOH, Fe(OH)_3_) shifts the system equilibrium, making the reduction of Fe^3+^ back to Fe^2+^ thermodynamically less favorable [13,15]; and (2) “kinetic shielding”, where these precipitates form a physical barrier on the catalyst surface, blocking active sites and drastically increasing the resistance to electron transfer, thereby kinetically hindering the Fe^3+^/Fe^2+^ conversion [31,44]. To counter these issues, strategies such as ligand complexation, surface defect engineering, and interface modulation have proven effective in mitigating precipitation and restoring rapid iron cycling (Table 1). These approaches collectively enhance both the efficiency of the iron cycle and the utilization of oxidants, providing crucial theoretical insights and practical strategies for deploying iron-based advanced oxidation systems in complex water environments. The following sections will elaborate on these enhancement strategies in detail.

## 3. Iron Cycle Enhancement Strategies and Their Mechanisms

### 3.1. Catalyst Design and Modification Strategies for Enhanced Iron Cycling

The strategic design of high-performance iron-based catalysts, as detailed in the following subsections, is intrinsically linked to their synthesis. The implementation of doping, heterointerfaces, defects, and morphological control is achieved through a suite of synthetic methods, each with its own merits and limitations [45]. Common techniques include (i) hydrothermal/solvothermal routes for high-crystallinity nanomaterials with defined morphologies, (ii) pyrolysis for carbon-based composites and single-atom catalysts, (iii) co-precipitation for scalable production of mixed metal oxides, and (iv) template-assisted methods for precise architectural control. The selection of a specific route (e.g., sol-gel, ball milling) is dictated by the target material’s properties, balancing factors such as scalability, cost, and the need for precise control over crystallinity, porosity, and elemental distribution [45,46].

#### 3.1.1. Electronic Structure Modulation via Elemental Doping

Elemental doping represents a sophisticated strategy to overcome the kinetic constraints of the Fe^3+^/Fe^2+^ cycle by deliberately engineering the electronic and coordination structures of iron-based catalysts (Figure 3). This strategy primarily operates through three distinct yet complementary mechanisms: (i) The incorporation of transition metals (e.g., Co, Cu, Mo) establishes coupled redox cycles where the dopant’s valence cycle (e.g., Co^2+^/Co^3+^) acts as an efficient electron-transfer mediator, shuttling electrons to Fe^3+^ and bypassing its slow direct reduction. (ii) Non-metal dopants (e.g., S, N, B) primarily modify the electronic structure of the support matrix (e.g., carbon), enhancing electron density and creating defined coordination environments to stabilize iron active sites and lower activation energies. (iii) Dual-element doping schemes synergize these effects, combining rapid metal redox cycling with an optimized electron-conducting matrix to further enhance performance and pathway selectivity. This approach introduces foreign elements—including both transition metals and non-metals—that actively participate in or facilitate electron transfer processes, thereby accelerating persulfate activation and sustaining catalytic longevity (Figure 4a) (Table 2).

The incorporation of transition metals (e.g., Co, Cu, Mo) establishes coupled redox systems wherein the dopant’s valence cycle (e.g., Co^2+^/Co^3+^, Cu^+^/Cu^2+^) acts as an efficient electron-transfer mediator. For instance, in Co-Fe_3_O_4_@FeOOH, the Co^2+^/Co^3+^ couple facilitates interfacial electron donation to Fe^3+^, enhancing the methylene blue degradation rate by 36-fold while restricting cobalt leaching to below 0.457 mg/L [47]. Beyond electron shuttling, certain dopants modulate the electronic properties of iron active sites. Density functional theory (DFT) calculations reveal that Mo doping in Fe/Mo@C downshifts the d-band center of iron, strengthening PMS adsorption and reducing the energy barrier for O–O bond cleavage which culminates in a 26-fold increase in bisphenol A degradation rate [48].

Non-metal dopants (e.g., S, N, B) primarily modify the support matrix to enhance electron density and create defined coordination environments (Figure 3). Electron-rich nitrogen configurations (e.g., pyrrolic N) in Fe-N-BC donate electron density to iron centers, stabilizing Fe^2+^ and lowering the activation energy for persulfate dissociation (ΔG = 23.54 kcal/mol), achieving 90.2% sulfamethoxazole degradation within 40 min [49]. Sulfur doping introduces reversible redox-active sites (e.g., S^2−^/S^0^) that directly reduce Fe^3+^, as demonstrated in Fe/S-NC, where thiophenic sulfur promotes interfacial electron polarization and sustains Fe^2+^ regeneration, maintaining 96.3% activity after five cycles [50].

Dual-element doping schemes yield synergistic effects by combining rapid metal redox cycling with enhanced electron transport through a modified carbon matrix. In FeCo@NBC, cobalt accelerates iron reduction while nitrogen doping increases graphitization (Iₚ/I_G = 1.04), collectively promoting a singlet oxygen-dominated pathway and high degradation efficiency for bisphenol S [51]. Crucially, dopant identity dictates reaction mechanism: sulfur incorporation in Fe@S_4_NC shifts the dominant reactive species from singlet oxygen to sulfate radicals, elevating sulfadiazine mineralization by 32% [52].

In essence, elemental doping enhances iron cycling through three convergent mechanisms: coupled redox mediation [47], electronic structure tailoring [48], and created defined coordination environments [49,50]. This strategy provides a foundational methodology for circumventing kinetic bottlenecks in iron-persulfate catalysis through targeted materials design. However, the choice of dopant is not arbitrary but should be guided by the desired electronic modulation and catalytic function. General principles for rational dopant selection are emerging from combined experimental and theoretical studies: (1) For enhancing electron transfer, transition metals with compatible and rapid redox cycles (e.g., Co^2+^/Co^3+^, Cu^+^/Cu^2+^) relative to Fe^3+^/Fe^2+^ are preferred. Their efficacy can be predicted by comparing standard reduction potentials and the energy barriers for electron transfer calculated via Density Functional Theory (DFT). (2) For modulating the electronic structure of the iron center or carbon support, non-metals with distinct electronegativities (e.g., N, S, B) are chosen to create electron-deficient or electron-rich regions, which can be tailored to optimize persulfate adsorption and activation. (3) The selection of dopant precursors is critical. Metal dopants often originate from nitrate, chloride, or acetylacetonate salts, chosen for their solubility and decomposition behavior during synthesis. Non-metal dopants are typically introduced from nitrogen-rich (e.g., urea, melamine), sulfur-rich (e.g., thiourea, L-cysteine), or boron-containing (e.g., boric acid) precursors. The selection is based on the precursor’s ability to incorporate efficiently into the host matrix at the synthesis temperature and its impact on the final material’s morphology and defect density. Ultimately, the integration of DFT calculations—predicting parameters like d-band center, adsorption energy, and charge density distribution—prior to experimentation is becoming an indispensable tool for the rational design of doped catalysts.

#### 3.1.2. Heterointerface Engineering for Enhanced Electron Transfer

Constructing heterostructures represents a sophisticated materials design strategy to overcome electron-transfer limitations in iron-based persulfate activation (Figure 4b). By deliberately engineering interfaces between iron compounds and carefully selected secondary phases—such as conductive carbons or other transition metal oxides—this approach creates synergistic effects that markedly accelerate the Fe^3+^/Fe^2+^ cycle and mitigate deactivation processes [53].

The integration of carbonaceous materials establishes efficient electron-transfer pathways that facilitate iron valence cycling. In core–shell Fe_3_O_4_@NCNTs, the graphitic carbon layers serve as conductive networks, tripling the rate of Fe^3+^ reduction while significantly suppressing iron leaching [54]. Confinement of Fe_2_O_3_ within oxidized carbon nanotubes (Fe_2_O_3_@OCNT) enhances electronic interaction and increases PMS adsorption, achieving 96.1% tetracycline degradation with minimal iron leaching (<10 μg/L) [55]. Biochar-based systems leverage inherent functional groups (e.g., C=O, pyridinic N) and defect sites to improve iron dispersion and electron shuttling. The synergistic effect between graphitic carbon and Fe–Nₓ sites in FeN@BC enables complete tetracycline degradation within 30 min across a broad pH range (3–9) [56].

Introducing a second metal species creates bimetallic redox pairs that establish efficient electron-relay mechanisms. In a Fe–Cu dual-atom catalyst, Cu^+^ serves as an electron shuttle—preferentially reducing Fe^3+^ while oxidizing persulfate—resulting in a 40% higher Fe^2+^ regeneration rate and a sixfold increase in PDS activation [57]. In Fe–Mn@biochar systems, Mn^3+^/Mn^4+^ cycles drive Fe^2+^ regeneration and contribute to 70% of radical generation, boosting the degradation rate constant by 20 times compared to monometallic catalysts [58].

Beyond enhancing electron transfer, heterostructures exert precise control over reaction pathway selection. Confinement effects and tailored interfaces can favor non-radical routes that are less susceptible to quenching by complex water matrices. For instance, Fe–N_4_ sites in single-atom catalysts activate PMS via direct electron transfer, while oxygen vacancy-rich ZnFe_2_O_4_/CNTs promote ^1^O_2_ generation with 89% contribution [20,59]. Encapsulation strategies, such as Fe_3_C encapsulated in graphitic carbon (Fe_3_C@C), reduce metal leaching by 90%, effectively decoupling high activity from stability losses [60].

In summary, heterointerface engineering enhances persulfate activation through three complementary mechanisms: the electronic pole, where conductive interfaces lower energy barriers for Fe^3+^ reduction; the synergistic pole, where bimetallic relays enable continuous electron shuttling; and the pathway pole, involving confinement-induced control over radical and non-radical pathways. This multi-level strategy provides a universal blueprint for designing robust iron-based catalysts with enhanced activity and stability. The rational design of such interfaces is paramount. The selection of secondary phases for constructing heterointerfaces follows a similar rationale driven by desired functionality. Conductive carbons (e.g., graphene, carbon nanotubes) are chosen to provide high-speed electron pathways, while secondary metal oxides or sulfides are selected to establish synergistic redox cycles or to create built-in electric fields that enhance charge separation. The choice is increasingly guided by theoretical predictions of interfacial energy, band alignment, and electron affinity at the junction, ensuring the designed interface facilitates the directional electron transfer crucial for Fe^3+^ reduction.

#### 3.1.3. Defect Engineering for Regulating Electron Transfer and Reaction Pathways

Defect engineering has emerged as a pivotal strategy for modulating the electronic structure of iron-based catalysts to overcome kinetic limitations in persulfate activation. By deliberately introducing vacancies—such as oxygen vacancies (Ov) and sulfur vacancies (Sv)—this approach optimizes the local coordination environment and electronic properties of metal active sites, thereby enhancing electron transfer efficiency, regulating reaction pathways, and improving catalytic stability (Figure 4c).

Oxygen vacancies function as electron reservoirs that significantly increase electron density at the catalyst surface. In CoWO_4−x_, Ov-induced localized electron rearrangement promotes direct electron donation to adsorbed PMS molecules, reducing the activation energy for O–O bond cleavage [22]. Similarly, graphene-supported α-Fe_2_O_3−x_ exhibits improved electrical conductivity due to Ov, which enhances Fe^2+^ regeneration and sustains persulfate activation [61]. Sulfur vacancies play a complementary role by exposing under-coordinated metal sites and optimizing adsorbate-catalyst interactions. In FeS/MoS_2_ systems, Sv exposes active Mo^4+^ sites that serve as electron mediators, facilitating Fe^3+^ reduction and promoting continuous iron cycling [23].

Beyond enhancing electron transfer, vacancy defects exert precise control over reaction mechanism orientation. Oxygen vacancies can lower the energy barrier for molecular oxygen adsorption, thereby promoting the generation of singlet oxygen (^1^O_2_) [62]. This is exemplified in MnCeO_x_–H, where Ov facilitates the formation of metastable PMS surface complexes that decompose preferentially via the ^1^O_2_ pathway [63]. Sulfur vacancies in CoFe_2_O_4−x_ mediate a stepwise reduction pathway, converting PMS into O_2_•^−^ and subsequently to ^1^O_2_ [64]. Low-coordination defect environments—such as asymmetric oxygen vacancies (As-OVs) in CuZnO_x_ [65] and Co-N_2_ sites in single-atom catalysts—further promote direct electron transfer processes (ETPs) by stabilizing metal–PMS* complexes with optimized orbital overlap [66].

Crucially, vacancy engineering addresses the persistent challenge of metal leaching through structural stabilization. Defects serve as structural anchors that stabilize metal centers via lattice confinement and electronic compensation. In CdxZn_1−x_S@BC, Zn-induced charge redistribution and vacancy formation suppress cadmium oxidation, reducing leaching by 62.8% [67]. Prolonged catalytic stability is achieved in silicate-confined systems, where iron cycling remains effective over 192 h [68]. The presence of Ov in FeVO_3−x_ [69] and Co_3_O_4_@C-500 [70] further stabilizes low-valence states (V^4+^, Co^2+^), enabling high reactivity over multiple cycles.

In summary, defect engineering enhances persulfate activation through three interconnected mechanisms: electronic facilitation, where vacancies act as electron donors/acceptors to accelerate metal redox cycling; pathway control, where defect-induced electronic restructuring favors specific oxidation routes (^1^O_2_ and ETP); and structural stabilization, whereby vacancies anchor metal species, suppress dissolution, and extend catalyst lifespan. This approach provides atomic-level insights for the rational design of highly efficient and stable iron-based catalytic systems.

#### 3.1.4. Morphological and Structural Engineering for Enhanced Mass and Electron Transport

Rational manipulation of catalyst morphology and structure plays a critical role in overcoming intrinsic limitations in persulfate activation by optimizing physical and chemical pathways for mass diffusion and electron transfer (Figure 4d). This approach enhances the accessibility of active sites, facilitates reactant and charge transport, and tailors reaction environments, thereby significantly improving the efficiency and sustainability of iron-based catalytic systems.

Nanostructural engineering increases specific surface area and exposes a greater density of active sites, improving PMS/PS adsorption and activation kinetics. For example, a SiO_2_-templated and NaNO_3_-activated porous carbon structure achieved a specific surface area of 773 m^2^/g. The abundant structural defects and synergistic FexOy-N-C effects enhanced PMS activation, enabling efficient pollutant degradation over a broad pH range (3–9) [71]. Hollow core–shell architectures, such as Fe_3_O_4_@CoFe_2_O_4_, utilize an internal cavity to increase reactive surface area while the outer shell protects the ZVI core from passivation, enabling sustained Fe^2+^ release and extending catalyst lifetime to 22 h [72].

Low-dimensional nanostructures—such as two-dimensional nanosheets and one-dimensional nanowires—shorten electron transport distances, thereby accelerating the Fe^2+^/Fe^3+^ cycle and improving oxidant utilization efficiency. Ultrathin Fe_3_O_4_ nanosheets (≈1 nm thickness) with abundant oxygen vacancies reduced the O–O bond cleavage energy barrier, increasing the PMS activation rate by five times [73]. MoS_2_ nanosheets coating Fe_3_O_4_ form Fe-S bridging interfaces that accelerate electron transfer and Fe^2+^ regeneration, achieving a degradation rate four times higher than that of pure Fe_3_O_4_ [21].

Ordered pore channels and hierarchical structures reduce mass transfer resistance and enhance the diffusion of pollutants and oxidants to active sites. Ordered porous Fe_3_O_4_@N–C cubes with acid-etched aligned channels (OA-P structure) accelerated ion/electron transport, simultaneously enhancing the iron cycle and PMS activation efficiency (99.8% BPA degradation within 60 min) [74]. Hollow flower-like γ-MnO_2_ loaded with iron was assembled from ultrathin nanosheets; the hierarchical porosity promoted As(III) diffusion to iron active sites, yielding a synergistic oxidation–adsorption capacity of 102.84 mg/g [75].

Certain morphological features further optimize catalytic performance through defect engineering or surface reconstruction, steering radical versus non-radical pathways. In MoS_2_-Fe, sulfur vacancies (Sᵥ) enhanced PMS adsorption and facilitated the formation of a FeIII–PMS* complex, driving non-radical pathways via ^1^O_2_ and Fe(IV)=O (contributing 45.5%) [76]. In core–shell Fe@C-1000, pore formation during Fe^0^ consumption allowed internal Fe_3_C to migrate to the surface, replenishing active sites with nearly zero iron leaching (0.08–0.23 mg/L) and shifting the dominant ROS from SO_4_•^−^ to high-valent iron Fe(IV) [77].

Morphological and structural regulation enhances persulfate activation through three fundamental mechanisms: improved mass transfer through hierarchical porosity, accelerated electron transport via low-dimensional architectures, and optimized reaction pathways through defect-assisted interface engineering. These strategies collectively address key limitations in iron-based persulfate activation and provide a foundation for designing high-performance catalytic systems.

In summary, the catalyst design strategies discussed in this section collectively address the iron cycling bottleneck through three fundamental mechanisms: electronic structure modulation, interface engineering, and mass/electron transport optimization. Elemental doping introduces foreign elements that create electron-transfer mediators and modify the coordination environment of iron sites, significantly enhancing the Fe^3+^/Fe^2+^ conversion kinetics [48,50]. However, the long-term stability of dopants and potential leaching issues require careful consideration. Heterointerface engineering constructs hybrid interfaces that provide continuous electron-transfer pathways and establish synergistic redox relays between different components [55,57]. While highly effective, the structural integrity of these interfaces under prolonged operation remains a challenge. Defect engineering creates vacancies that serve as electron reservoirs and modulate surface adsorption/activation properties, enabling precise control over reaction pathways [22,67]. Nevertheless, the controlled generation and maintenance of these defects under realistic conditions need further investigation. Morphological engineering optimizes physical structures to enhance mass diffusion and active site accessibility, though scalability and practical implementation of complex nanostructures pose significant hurdles [73,74]. Looking forward, the integration of multiple strategies—such as doped materials with tailored defects and controlled morphology—represents a promising direction for developing robust, high-performance iron-based catalysts that maintain efficiency under environmentally relevant conditions.

**Table 2 nanomaterials-15-01712-t002:** Comparative Analysis of Catalyst Design Strategies for Enhancing Iron Cycling in Persulfate Activation.

Strategy	Primary Mechanism & Electronic Modulation	Key Structural Feature	Dominant Enhancement in Iron Cycling	Typical Performance Outcome	References
Elemental Doping	Introduces foreign elements to modify the electronic structure and create electron-transfer mediators	Atomic-level dispersion of dopants; formation of active sites	Couples Fe^3+^/Fe^2+^ with a faster redox pair; lowers activation energy for Fe^3+^ reduction.	26-fold increase in BPA degradation rate; >96% activity retention after 5 cycles.	[48,50]
Heterointerface Engineering	Constructs hybrid interfaces to provide continuous electron-transfer pathways and synergistic redox relays.	Core–shell structures; bimetallic junctions; carbon-supported composites.	Accelerates interfacial electron transfer from support or second metal to Fe^3+^.	6-fold higher PDS activation; 96% degradation with minimal leaching (<10 μg/L).	[55,57]
Defect Engineering	Creates vacancies (Oᵥ, Sᵥ) that act as electron reservoirs and modulate surface adsorption/activation.	Oxygen/sulfur vacancies; under-coordinated metal sites; asymmetric coordination.	Serves as localized electron donor for Fe^3+^ reduction; stabilizes low-valence states.	62.8% reduction in metal leaching; sustained activity over 192 h.	[22,67]
Morphological Engineering	Controls physical structure to shorten mass/electron transport distances and increase active site exposure.	Hollow structures; ordered pores; 2D nanosheets; hierarchical porosity.	Improves diffusion of reactants to sites and electrons to Fe centers.	5-fold higher PMS activation rate; 99.8% degradation in 60 min.	[73,74]

### 3.2. Reaction System Optimization Strategies

Building upon the intrinsic strategies of catalyst design discussed in Section 3.1—which focused on permanently embedding enhanced properties within the catalyst’s architecture—this section shifts the paradigm to extrinsic system optimization. Here, the enhancement of the iron cycle is achieved not by altering the catalyst’s fundamental identity, but by strategically engineering the external reaction environment. This is accomplished through the introduction of soluble chemical agents or solid supplements that act as auxiliary components within the reaction system. These extrinsic agents work in concert with the catalyst, operating alongside it in the solution or at the interface to dynamically supply electrons, mediate charge transfer, and prevent deactivation. This approach offers a complementary pathway to overcome kinetic limitations, often providing greater operational flexibility and rapid efficacy, albeit sometimes at the cost of long-term reagent consumption or potential secondary effects.

#### 3.2.1. Introduction of Chemical Reductants

The introduction of chemical reductants addresses the intrinsic kinetic limitation of the Fe(III)/Fe(II) cycle by providing exogenous electrons, thereby enabling sustained persulfate activation and dramatically enhancing the degradation efficiency of refractory pollutants (Figure 5a). This strategy operates through direct electron donation, complexation-mediated reduction, or the formation of transient reactive intermediates that facilitate iron valence cycling (Table 3). Unlike the permanent electronic structure modulation achieved by doping, chemical reductants function as soluble, sacrificial electron donors in the solution phase, providing a transient but powerful boost to the iron cycle.

Inorganic reductants, such as hydroxylamine (HA) and dithionite (DTN), primarily function via direct electron transfer to iron centers. HA enhances atrazine degradation by 38-fold in a Fe_3_O_4_/PMS system, serving both as a homogeneous activator and a surface-mediated electron shuttle through Fe–O–N bonding [12]. DTN exhibits stronger reducibility by generating SO_4_•^−^ radicals, which drive a coupled vanadium-iron redox cycle in FeVO_4_/PMS systems, achieving >90% carbamazepine removal within 300 s (Lai et al., 2021 [78]). Sulfite-based reductants (e.g., HSO_3_^−^) demonstrate remarkable pH adaptability through complexation with dissolved iron, enhancing reactivity via multiple pathways: reduction of Fe(III), reaction with Fe(IV) to produce SO_4_•^−^, and in situ generation of PMS [12].

Organic reductants offer the dual function of reducing Fe(III) and solubilizing iron species, thereby preventing precipitation and maintaining iron bioavailability. Ascorbic acid (AA) significantly enhances sulfamethazine degradation (72.4% improvement) in Fe(III)/PDS systems by reducing surface-bound Fe(III) and forming soluble [Si–Fe(II)] complexes, minimizing off-pathway Fe(II) oxidation (Li et al., 2023 [79]). Cysteine (Cys) acts as both chelator and reductant, forming soluble Fe(III)-complexes that remain active across pH 3–9, increasing the quinolinic acid degradation rate sixfold [80]. Oxalic acid (OA) operates indirectly by accelerating auxiliary metal cycles—e.g., promoting V(V)/V(IV) cycling in VO_2_/H_2_O_2_ systems—which in turn facilitates Fe(III) reduction [81].

Beyond accelerating iron cycling, chemical reductants can modulate reactive species distribution. HA shifts the dominant pathway in Fe(II)/PDS systems from high-valent iron (Fe(IV)) to radical-based routes (SO_4_•^−^/•OH) by rapidly reducing Fe(IV) and regenerating Fe(II) [82]. In contrast, DTN enhances electron utilization in multi-metal systems via vanadium-mediated cycling, minimizing radical quenching and improving oxidant efficiency [78].

In summary, chemical reductants enhance iron-persulfate systems through three core mechanisms: thermodynamic facilitation of Fe(III) reduction, kinetic enhancement of electron transfer, and morphological control of iron speciation. This approach provides a versatile strategy for overcoming the kinetic limitations of iron-based persulfate activation across a wide range of environmental conditions.

#### 3.2.2. Electron Shuttles for Mediated Redox Cycling

The introduction of electron shuttles establishes a reversible electron-transfer system that decouples persulfate activation from the slow kinetics of heterogeneous iron reduction. This strategy employs redox-active mediators capable of cycling between oxidized and reduced states, thereby facilitating continuous Fe(III) reduction and sustaining high catalytic activity without permanent consumption (Figure 5b). In contrast to the static, conductive interfaces created by heteroengineering, electron shuttles operate through dynamic, reversible redox reactions in the bulk solution, mediating electron transfer over distance.

Quinoid compounds function as highly efficient molecular shuttles due to their rapid two-electron redox cycling between quinone (electron-accepting) and hydroquinone (electron-donating) states. In methyl-p-benzoquinone (MBQ)-enhanced systems, this shuttle mechanism operates through precisely coordinated steps: MBQ accepts electrons from electron-rich domains to form methyl-hydroquinone (MHQ), which subsequently reduces Fe(III) to Fe(II). This cyclic electron transfer enables exceptional atrazine degradation across pH 2–7, significantly outperforming conventional Fe(II)/PDS systems [83]. The universal nature of this mechanism is demonstrated by benzoquinone (BQ), which elevates atrazine degradation from 23.2% to 99.8% in Fe(III)/PMS systems through analogous electron-shuttling pathways.

Natural humic substances represent evolutionarily optimized electron shuttles that integrate quinoid moieties with metal-complexing functional groups. In HA-Fe@BC/PDS systems, the quinone groups in humic acid accept electrons from the conductive biochar matrix to form persistent semiquinone radicals, which drive efficient Fe(III) reduction and enhance rhodamine B degradation kinetics by eightfold [81]. When coupled with FeAl-layered double hydroxides, dissolved organic matter (DOM) not only facilitates electron transfer but also generates oxygen vacancies that stabilize Fe(II) species. While humic acids mitigate anion interference (Cl^−^/NO_3_^−^) through complexation, their potential quenching effects at high concentrations necessitate careful dosage optimization.

Sulfur-based compounds introduce advanced electron transfer capabilities through multi-electron redox processes. In the SC-NiFe_2_S_4_/PMS system, surface sulfide species (S^2−^) enable concurrent Ni^2+^/Ni^3+^ and Fe^2+^/Fe^3+^ cycling while creating oxygen vacancies that synergistically activate PMS through a predominant singlet oxygen (^1^O_2_) pathway [84]. More remarkably, dithiothreitol (DTT) in FeVO_4_/PMS systems generates strongly reducing SO_2_•^−^ radicals that initiate a cascade electron transfer: reducing V^5+^ to V^4+^, which then transfers electrons to Fe(III) through Fe–O–V bonds. This sophisticated electron relay system maintains >90% degradation efficiency over multiple cycles while minimizing metal leaching [78].

The strategic integration of electron shuttles represents a paradigm shift in persulfate activation, moving from direct activation to mediated electron transfer regimes. These molecular mediators not only accelerate iron cycling kinetics but also enable unprecedented control over reaction pathways and selectivity, offering a robust strategy for overcoming kinetic limitations in complex environmental matrices.

#### 3.2.3. Elemental Substances as Reductive Modulators

The incorporation of elemental substances—both metallic and nonmetallic—offers a robust strategy to overcome electron-transfer limitations in iron-based persulfate systems. This approach fundamentally alters the local electronic environment and interfacial reaction dynamics, enabling sustained iron cycling through direct electron donation, structural modification, and the creation of synergistic redox partnerships (Figure 5c). This strategy differs from the morphological engineering of catalysts, as it involves the addition of bulk elemental materials (e.g., ZVAl) as co-reagents that corrode to provide electrons, rather than structuring the iron catalyst itself.

Metallic elements such as zero-valent aluminum (ZVAl) and copper function as direct electron donors or co-catalytic mediators. ZVAl and its corrosion products serve dual roles: activating PMS through direct electron transfer to generate SO_4_•^−^, and concentrating pollutants near reactive sites through adsorption, thereby enhancing local reaction efficiency [85]. Copper enhances iron cycling within bimetallic frameworks (e.g., CuFeBDC); the Cu^+^/Cu^2+^ cycle rapidly reduces Fe^3+^, while Cu^+^ is regenerated through oxidation by PMS, establishing an autocatalytic loop [86]. Zinc doping modifies the electronic structure of iron active sites, lowering the redox potential of the Fe^3+^/Fe^2+^ couple and increasing electron density, thereby significantly accelerating Fe^3+^ reduction [87].

Nonmetallic elements, particularly boron and carbon, operate through distinct yet complementary mechanisms. Amorphous boron acts as a potent reducing agent at the Fe@NC/B interface, continuously supplying electrons to iron species to drive Fe^3+^ reduction and potentially directly activating PMS to form both SO_4_•^−^ and ^1^O_2_ (Wang et al., 2023 [88]). Carbon layers simultaneously function as physical barriers (e.g., in carbon-coated nZVI), effectively preventing oxidative passivation of iron nanoparticles and enhancing long-term stability [29]. Furthermore, specific carbon architectures can induce non-radical pathways, degrading contaminants via direct electron transfer or singlet oxygen generation.

The strategic integration of these elements enables precise control over reaction mechanisms and iron speciation. Elemental additives optimize the local electronic environment of iron active sites—metallic dopants introduce new energy levels and reduce charge transfer resistance, while nonmetallic elements modify work function and surface electron density. These modifications collectively lower the activation energy for PMS decomposition and promote the regeneration of active Fe^2+^ species. Additionally, elemental enhancers improve catalyst resilience in complex matrices: carbon coatings mitigate interference from common anions (e.g., Cl^−^, CO_3_^2−^) and natural organic matter, while metallic reductants like ZVAl maintain activity over a broader pH range.

In essence, the use of elemental substances provides a multi-faceted approach to breaking the kinetic constraints of iron cycling. Through electron donation, interface engineering, and structural stabilization, these elements work in concert to sustain efficient persulfate activation, offering a versatile and effective strategy for environmental remediation applications.

In critically evaluating these system optimization strategies, it becomes evident that each approach represents a distinct compromise between electron transfer efficiency and practical implementability. Chemical reductants achieve the most rapid Fe^3+^ reduction kinetics through direct electron transfer, yet this advantage is counterbalanced by their stoichiometric consumption and potential secondary contamination—fundamental limitations that question their environmental sustainability. Electron shuttles offer an elegant catalytic alternative through reversible redox cycling, but their molecular-level operation renders them vulnerable to matrix interference and degradation, revealing a critical sensitivity to environmental conditions that challenges field application. Elemental substances provide the most robust electron donation platforms through bulk-phase interfaces, yet their macroscopic nature introduces equally macroscopic challenges of metal leaching and surface passivation that ultimately limit longevity. This critical analysis reveals an unresolved tension in the field: the strategies that achieve the fastest iron cycling kinetics tend to be the least sustainable, while the most environmentally compatible approaches often suffer from kinetic limitations. Future research must therefore move beyond optimizing individual strategies and instead focus on developing integrated systems that combine the rapid kinetics of chemical reductants with the sustainability of catalytic mediators and the stability of solid-phase donors, while addressing the fundamental mechanistic trade-offs identified here.

**Table 3 nanomaterials-15-01712-t003:** Comparative Analysis of System Optimization Strategies via Reductive Enhancement.

Strategy	Core Mechanism & Action Scale	Primary Role in Iron Cycling	Key Advantage	Inherent Limitation/Risk	References
Chemical Reductants	Homogeneous molecular-scale reduction. Provides exogenous electrons via direct donation (e.g., HA, DTN) or complexation-mediated transfer (e.g., AA, Cys).	Overcomes kinetic barrier of Fe(III) reduction. Acts as a soluble electron source to rapidly regenerate Fe(II).	Immediate efficacy; Broad availability; Can shift reaction pathways (e.g., from Fe(IV) to radicals).	Consumable; Requires continuous dosing; Potential formation of harmful byproducts; Increases operational cost.	[12,78]
Electron Shuttles	Reversible molecular-mediated transfer. Establishes a catalytic cycle where mediators (e.g., quinones, humics) oscillate between redox states.	Decouples Fe(III) reduction from oxidant activation. Enables continuous electron transfer without mediator consumption.	Catalytic (non-consumable); High selectivity; Mitigates anion interference; Operates over a wide pH range.	Susceptible to radical attack; Performance pH-dependent for natural mediators; Cost barriers for synthetic analogs.	[81,83]
Elemental Substances	Heterogeneous interface & bulk electron donation. Metallic elements (e.g., ZVAl, Cu^0^) donate electrons; Carbon matrices act as conductive mediators.	Provides a sustained electron flux. Serves as a bulk electron reservoir or creates conductive networks for electron transfer.	Sustained release of electrons; Multifunctionality (e.g., adsorption, pH buffering); ZVAl offers a wide pH applicability.	Potential metal leaching (e.g., Al^3+^); Passivation of metal particles over time; Slower initial kinetics compared to molecular reductants.	[85,88]

### 3.3. External Energy/Field Assistance Strategies

#### 3.3.1. Photo-Assistance for Renewable Electron Injection

Light irradiation serves as a powerful external energy input to overcome the kinetic limitations of iron-mediated persulfate activation by providing a renewable source of electrons and holes (Figure 6a). This strategy fundamentally alters the reaction kinetics and pathways through photogenerated charge carriers that directly drive the critical Fe^3+^/Fe^2+^ cycle and facilitate advanced oxidation processes beyond thermal activation (Table 4).

The core mechanism involves photonic excitation that generates electron-hole pairs, enabling direct photoreduction of Fe^3+^ to Fe^2+^. This process illustrates how visible light irradiation on an Fe/Mn-Prussian blue analog@MXene composite injects photogenerated electrons into the dual metal-cycle pathways. This direct photoreduction bypasses the conventional kinetic barrier of iron reduction, where the corresponding performance data shows the system achieving 93.4% tetracycline degradation [89]. Beyond accelerating iron cycling, light facilitates the formation of multiple reactive species through both radical and non-radical pathways. In UV/CoFe_2_O_4_/periodate systems, photoactivation enables simultaneous generation of six distinct radical species, leading to complete pollutant degradation within 12 min [89].

Advanced material design strategies maximize light absorption and charge separation efficiency. Oxygen vacancy-rich ZnFe_2_O_4_ (ZFOᵥ) utilizes vacancies as electron reservoirs that localize electrons for persulfate activation, increasing degradation efficiency by 1.23-fold [90]. The construction of visible-light-responsive heterojunctions (e.g., MoS_2_/δ-FeOOH) extends absorption spectra while suppressing charge recombination through built-in electric fields, enabling nearly 100% RhB degradation within 35 min [91]. Defect engineering further enhances performance, as demonstrated by Se vacancies in CoSe_2_/FeSe_2_@CNT that modify the electronic structure to favor Fe^3+^ reduction, increasing radical yield by 3.8 times [92].

Photo-assisted systems exhibit remarkable adaptability to complex environmental conditions. The visible light/gallic acid/Fe^3+^/periodate system maintains high activity across pH 3–9, achieving complete BPA removal in 30 min under neutral conditions [93]. Natural mineral-based systems such as chalcopyrite/PAA/visible light retain 95% sulfamethoxazole degradation efficiency even in the presence of common anions (Cl^−^/HCO_3_^−^) [94]. The integration of photo-catalytic processes with other functions demonstrates multifunctional potential, as shown in FeMn-PBA@MXene systems that simultaneously degrade tetracycline (98.9%) and desalinate water, reducing salinity from 4000 ppm to 240 ppm [95].

Both radical and non-radical pathways are enhanced under light irradiation. In CoFe_2_O_4_/PMS, SO_4_•^−^ contributes 76% to degradation, though intermediate toxicity necessitates pathway control [96]. Non-radical routes such as ^1^O_2_ generation dominate in certain heterojunctions (e.g., ZnFe_2_O_4_/Bi_2_O_2_CO_3_/BiOBr), exceeding the contribution of SO_4_•^−^ [97]. High-valent iron species (e.g., Fe(IV)=O) also become more accessible under light, showing superior selectivity in contaminant degradation [98]. DFT simulations provide mechanistic validation, confirming that Fe-modified surfaces significantly reduce PMS adsorption energy and decrease activation barriers [99].

In summary, photo-assistance enhances iron-persulfate systems through three fundamental mechanisms: renewable electron injection for Fe^3+^/Fe^2+^ cycling, expanded reactive species diversity through photogenerated carriers, and improved environmental adaptability through multi-pathway activation. This approach represents a promising direction for developing sustainable and efficient water treatment technologies that leverage solar energy as a primary driving force.

#### 3.3.2. Thermal Assistance for Overcoming Activation Energy Barriers

Thermal energy activation represents a fundamental strategy for enhancing persulfate-based advanced oxidation processes by providing the necessary energy to overcome intrinsic activation barriers (Figure 6b). This approach operates through two synergistic mechanisms: direct facilitation of O–O bond cleavage in persulfate molecules and thermodynamic optimization of the iron redox cycle, thereby addressing both oxidant activation and catalyst regeneration limitations (Table 4).

The homolytic cleavage of the O–O bond in persulfates requires substantial energy input (Eₐ > 90 kJ/mol). Thermal energy directly supplies this activation energy, enabling efficient persulfate decomposition even in the absence of catalysts. For instance, the thermal activation of PDS for difloxacin degradation exhibits an Eₐ of 91.25 kJ/mol [100], significantly higher than that of Fe^2+^-activated systems (50–70 kJ/mol), underscoring the rate-limiting nature of this step. Beyond direct activation, thermal treatment tunes the iron electronic environment. Vacuum annealing (150–300 °C) increases the Fe^2+^/Fe^3+^ ratio in MIL-101(Fe) by up to 50%, thereby optimizing the alignment between the Fe^3+^/Fe^2+^ redox potential (E^0^ ≈ 0.77 V) and that of PMS (HSO_5_^−^/SO_4_^2−^, E^0^ = 1.82 V). This electronic optimization enhances the electron transfer rate constant by 2.3 times [101], significantly accelerating the iron-mediated activation cycle. However, this technique also presents significant limitations including equipment complexity, high energy consumption for maintaining vacuum conditions, scalability challenges for industrial applications, potential structural damage from prolonged treatment, and limitation to solid catalyst systems only [101].

The iron-mediated activation follows a chain reaction mechanism where thermal energy alleviates the kinetic bottleneck through two complementary pathways. First, it reduces the activation energy of Fe^3+^ reduction—as demonstrated in a coal/Fe_2_O_3_ system where Eₐ decreased from 62.79 kJ/mol to 56.50 kJ/mol at 60 °C, increasing the reduction rate threefold [102]. Second, autothermal reinforcement establishes a self-sustaining cycle whereby the exothermic nature of the Fe^2+^-PMS reaction (ΔH = −198 kJ/mol) maintains elevated temperature and promotes continuous radical generation, boosting radical yield by 3–5 times compared to non-thermal systems. The dominance of SO_4_•^−^ (>70%) under such thermal activation has been confirmed via ESR in systems including NO removal and H_2_S oxidation [103].

The integration of solar energy introduces a renewable pathway for thermal activation, combining photonic and thermal effects for enhanced efficiency. Solar irradiation provides both electronic excitation and thermal energy, enabling multi-field synergies. For example, UV-induced ligand-to-metal charge transfer (LMCT) in Fe^3+^-carboxylate complexes enhances the quantum yield of Fe^3+^ reduction, while the thermal component (>40 °C) facilitates PMS activation. In a solar/Fe^2+^/PMS system, the E. coli inactivation rate constant reaches 0.23 min^−1^—10 times higher than photolysis alone [104]. Low-grade solar-thermal energy (∼60 °C) can efficiently activate PDS for antibiotic degradation, even overcoming an Eₐ of ∼91.25 kJ/mol [100].

Thermal assistance demonstrates remarkable adaptability to various catalyst architectures and environmental conditions. The maintained elevated temperature prevents iron precipitation and stabilizes reactive intermediates across a broader pH range than conventional systems. Furthermore, the reduced reliance on precise catalyst design makes thermal activation particularly suitable for complex wastewater matrices where catalyst poisoning typically occurs.

In essence, thermal enhancement operates through dual fundamental mechanisms: thermodynamic facilitation of persulfate bond cleavage and kinetic acceleration of the iron redox cycle. This approach provides a versatile and often essential strategy for achieving efficient pollutant degradation, particularly for recalcitrant compounds requiring high activation energies. The integration with renewable solar-thermal energy further positions thermal activation as a sustainable pathway for advanced oxidation processes.

#### 3.3.3. Electrocatalytic Assistance for Controlled Iron Redox Cycling

Electrocatalytic activation represents a sophisticated strategy to overcome the kinetic limitations of iron-based persulfate systems through precise electronic control (Figure 6c). By applying an external potential, this method enables directional electron transfer and dynamic regulation of iron speciation, achieving sustained Fe^2+^ regeneration and significantly enhanced catalytic efficiency without chemical consumables (Table 4).

The fundamental advantage of electrochemical assistance lies in its ability to promote the cathodic reduction of Fe^3+^ to Fe^2+^, thereby breaking the rate-limiting step in traditional Fenton-like systems. In electro/Fe^3+^/PDS systems, cathodic electron transfer efficiently regenerates Fe^2+^, which activates persulfate to generate sulfate radicals (SO_4_•^−^) and hydroxyl radicals (•OH), enabling effective degradation of recalcitrant pollutants such as clofibric acid [105]. Heterogeneous catalysts like Fe-B further enhance the electron transfer kinetics between Fe^3+^ and Fe^2+^ under an electric field, considerably improving PMS activation efficiency [106]. This electrochemical acceleration addresses the intrinsic slow kinetics that plague conventional iron cycling systems.

Different iron species exhibit distinct activation behaviors under electrochemical promotion. Zero-valent iron (ZVI) serves dual roles as both a Fe^2+^ source and electron donor in ZVI-E-Fenton-PMS systems: ZVI undergoes anodic oxidative dissolution to release Fe^2+^, while the cathode simultaneously reduces Fe^3+^, creating a synergistic effect that promotes the formation of multiple radical species including •OH, SO_4_•^−^, and ^1^O_2_ [107]. For crystalline iron minerals such as pyrite (FeS_2_), electrochemical assistance leverages surface sulfide (S^2−^) as an electron mediator to facilitate Fe^2+^ regeneration. Simultaneously, sulfur intermediates (e.g., Sₙ^2−^, S^0^) enhance radical production, revealing a novel mineral-electric field synergistic mechanism [108].

Advanced electrode and catalyst designs have optimized iron cycling efficiency and system sustainability. Self-sustaining cathode materials (e.g., Fe_3_O_4_-CaO_2_) utilize the chemical cycle of CaO_2_/Ca(OH)_2_ to accelerate Fe^3+^ reduction, reducing energy consumption to 5% of that in conventional systems [109]. The introduction of atomic hydrogen (H*) via Pd/Al_2_O_3_ particle electrodes offers an efficient pathway for reducing Fe^3+^ while significantly minimizing iron sludge formation [110]. These systems perform effectively under near-neutral conditions (pH ~6.0) via heterogeneous iron (oxyhydr)oxides, overcoming the strong acidity requirement of traditional Fenton reactions [111].

Electrocatalytic activation provides exceptional control over reaction mechanisms, enabling a deliberate shift from radical to non-radical pathways to reduce scavenging effects and improve selectivity. In the E/N-S@Fe-PBC/PMS system, the combination of an electric field and nitrogen-sulfur co-doped biochar significantly promotes the generation of singlet oxygen (^1^O_2_), contributing up to 37% to pollutant degradation [112]. Electrode material selection profoundly influences the activation route: BDD anodes favor direct electron transfer to produce SO_4_•^−^, while DSA anodes promote surface-mediated non-radical oxidation [113]. This tunability provides a strategic advantage for treating complex matrices with varying compositions.

Despite these promising features, challenges remain in electrode cost, interference from co-existing substances, and long-term operational stability. Future research should focus on developing robust and affordable electrode materials, elucidating interfacial iron cycling mechanisms through in situ techniques, and demonstrating scalability in real-world environments. The integration of electrocatalytic processes with renewable energy sources further represents a critical pathway toward sustainable water treatment applications.

#### 3.3.4. Ultrasonic Assistance for Enhanced Interfacial Processes

Ultrasound irradiation enhances iron-based persulfate activation through cavitation-induced physical and chemical effects that fundamentally alter interfacial processes (Figure 6d). This approach operates through three primary mechanisms: microjet impingement that removes passivation layers, enhanced mass transfer that accelerates reactant availability, and sonochemical generation of reactive species that complement persulfate activation (Table 4).

The collapse of cavitation bubbles generates extreme local conditions (~5000 K, ~1000 bar) and powerful microjets that mechanically disrupt passivating iron oxide layers. In zero-valent iron (ZVI) systems, this continuous surface renewal facilitates the corrosion of Fe^0^ to release Fe^2+^, which activates persulfates to generate SO_4_•^−^ [114]. This process significantly improves electron transfer from Fe^0^ to PMS, enabling rapid degradation—99.76% removal of RhB within 12 min—while simultaneously promoting the reduction of Fe^3+^ back to Fe^2+^ [115]. The mechanical energy input alleviates Fe^3+^ accumulation and prevents reaction stagnation, overcoming a fundamental limitation of conventional homogeneous systems.

In heterogeneous catalytic systems, ultrasound enhances iron cycling through synergistic redox mechanisms. In FeS-based catalysts, S^2−^ oxidation to polysulfides (S_x_^2−^) releases electrons that drive Fe^3+^ reduction to Fe^2+^. The regenerated Fe^2+^ then activates persulfate to produce radicals, establishing a self-sustaining cycle [116]. This mechanism remains highly efficient under neutral conditions, achieving 86.6% degradation of diclofenac and overcoming the pH limitations of homogeneous Fenton systems [117]. Material architecture plays a crucial role: porous sponge iron (Fe^1^), with its high surface area, provides abundant Fe^2+^/Fe^3+^ sites and exhibits higher iron transformation rates than powdered iron (Fe^0^), significantly enhancing sludge dewatering (reducing CST by 89%) and soluble organic matter degradation [118].

Ultrasound assistance maintains long-term catalytic activity by countering deactivation processes. Cavitation microjets continuously remove passive oxide layers from material surfaces such as Fe_3_O_4_ or ZVI, exposing fresh active sites [119]. Support materials contribute to stability; in Fe_3_O_4_@biomass porous carbon (BPC), the carbon carrier disperses iron particles, reduces agglomeration, and preserves reactive interfaces, maintaining 70% norfloxacin degradation after four cycles [120]. Additionally, ultrasound induces mechanochemical effects that rearrange surface metal valences. For example, in Fe-NiOx catalysts, sonication exposes more high-valence metal sites (Ni^3+^/Fe^4+^), thereby improving PMS activation efficiency [121].

The integration of ultrasound with persulfate activation demonstrates particular effectiveness in complex matrices where conventional systems fail. The physical cleaning action mitigates fouling from natural organic matter, while the enhanced mixing overcomes diffusion limitations in high-viscosity environments. Furthermore, the simultaneous production of hydroxyl radicals from water sonolysis creates complementary oxidation pathways that enhance overall treatment efficiency.

Ultrasonic assistance enhances iron-persulfate systems through three convergent mechanisms: surface renewal via cavitational cleaning, enhanced mass and electron transfer through turbulent mixing, and sonochemical radical generation. This multi-mechanism approach provides a robust strategy for maintaining catalytic activity, preventing deactivation, and enabling operation under environmentally relevant conditions.

In critically assessing external field-assisted iron cycling, fundamental distinctions emerge in how different energy inputs activate and sustain the Fe^3+^/Fe^2+^ redox cycle. Photo-assistance drives charge separation and electron injection through photon absorption, enabling renewable Fe^3+^ reduction but facing intrinsic limitations in light penetration and quantum efficiency [89,90]. Thermal activation provides the requisite energy for persulfate homolysis and accelerates electron transfer kinetics, yet the substantial energy input raises concerns for sustainable implementation [100,101]. Electrochemical systems achieve precise control over iron speciation through applied potentials, enabling directed electron transfer without chemical additives but confronting challenges of electrode stability and mass transport limitations [105,109]. Ultrasonic activation uniquely combines physical cleaning of passivated surfaces with enhanced mass transfer through cavitation effects, though energy intensity and reactor scalability remain significant barriers [114,115]. Collectively, these approaches reveal a critical trade-off: external energy inputs can dramatically enhance iron cycling kinetics but often at the cost of operational complexity and energy consumption that may limit practical application. Future advances should focus on developing energy-adaptive systems that optimize the synergy between multiple activation mechanisms while minimizing external energy requirements through intelligent catalyst design and process integration.

**Table 4 nanomaterials-15-01712-t004:** Comparative Analysis of External Field Assistance Strategies for Iron Cycling Enhancement.

Strategy	Fundamental Principle & Action Scale	Primary Role in Iron Cycling	Unique Advantage	Practical Consideration & Limitation	References
Photo-Assistance	Photon-electron conversion. Utilizes photonic energy to generate electron-hole pairs for Fe^3+^ photoreduction and radical generation.	Provides renewable electrons; Expands reactive species diversity (e.g., via LMCT).	Solar energy utilization; Enables multi-pathway activation; High tunability via material design.	Dependent on light penetration; Catalyst requires photoactivity; Possible light shielding in complex matrices.	[90,95]
Thermal Assistance	Thermal energy input. Overcomes activation energy barriers for O–O bond cleavage and accelerates reaction kinetics.	Lowers activation energy for both persulfate decomposition and Fe^3+^ reduction; Enables autothermal cycles.	Universal applicability; No need for specialized catalysts; Effective for recalcitrant compounds.	High energy consumption; Limited control over reaction pathways; Potential for byproduct formation.	[100,101]
Electrocatalytic Assistance	Precise electron delivery. Applies potential to directionally drive electrons to Fe^3+^ at the cathode interface.	Sustains controlled Fe^2+^ regeneration via direct cathodic reduction; Decouples oxidation and reduction sites.	Precise redox control; Tunable reaction mechanisms; Minimal chemical consumption.	Electrode cost and fouling; Mass transfer limitations; System complexity and scalability.	[105,109]
Ultrasonic Assistance	Cavitation-induced phenomena. Uses acoustic cavitation for surface cleaning, enhanced mass transfer, and sonochemistry.	Prevents passivation via surface renewal; Enhances mass transfer of reactants to active sites.	Operates under any pH; Mitigates catalyst fouling; Effective in viscous or particulate-laden systems.	High energy intensity; Limited reactor design scalability; Potential for equipment erosion.	[114,121]

## 4. Application and Performance Evaluation in Environmental Remediation

### 4.1. Mechanism-Tailored Strategies for Different Contaminant Classes

The deployment of iron-based persulfate systems for environmental remediation necessitates a mechanistic understanding of how enhanced iron cycling translates into effective pollutant degradation across diverse contaminant classes. Performance evaluation must extend beyond simple degradation metrics to encompass pathway selectivity, matrix effects, and long-term sustainability, establishing a critical link between fundamental mechanisms and practical application.

Molecular architecture dictates contaminant susceptibility to specific oxidative pathways, requiring strategic alignment between iron cycling enhancement and target pollutant characteristics. Electron-rich compounds, including phenolics and dyes, undergo preferential degradation through non-radical pathways (e.g., direct electron transfer or singlet oxygen), as their high electron density favors selective oxidation (Figure 7a). Crucially, the oxidative pathway shifts for contaminants where the primary challenge is not initial electron abstraction, but the cleavage of inherently stable molecular frameworks. However, for refractory aromatic compounds, the sustained generation of SO_4_•^−^ and •OH proves essential for forceful aromatic ring cleavage and complete antibiotic inactivation [122]. For electron-withdrawing contaminants such as benzoic acid and nitrobenzene, non-radical pathways (e.g., ^1^O_2_ or surface electron transfer) often exhibit limited efficacy. This is primarily due to the reduced electron density of the pollutant’s aromatic ring, which hinders its interaction with mild oxidants. Bimetallic systems establish coupled redox cycles that bypass kinetic limitations of solitary iron cycling: in Fe/Mo@C, Mo doping stabilizes lower iron valence states while modulating the d-band center of Fe active sites, enhancing PDS activation through facilitated O–O bond cleavage and achieving a 26-fold increase in sulfamethoxazole degradation rate [48]. Halogenated contaminants present distinct challenges requiring specialized approaches. While pesticides undergo radical-mediated dehalogenation, Halogenated Substituted Benzene resist conventional oxidation due to exceptional C–X bond stability (Figure 7c). Sulfur-mediated strategies address this dichotomy through enhanced iron cycling; surface S^2−^ species in FeS@biochar provide electrons for Fe(III) reduction, doubling iron turnover rates for 95.9% 2,4-D degradation [123].

While conventional radical pathways often prove inadequate for PFAS degradation due to radical scavenging and the exceptional stability of C-F bonds, iron-activated persulfate systems achieve effective PFOA removal through integrated mechanisms. Research demonstrates that zero-valent iron (ZVI) synergistically activates persulfate by generating sulfate radicals while simultaneously facilitating PFOA decomposition, achieving 67.6% degradation at 90 °C [124]. More significantly, heterogeneous systems employing FeS overcome the limitations of homogeneous radical reactions by concentrating PFOA at the mineral surface (KD = 100–500 L/kg), creating a localized reaction environment that minimizes interference from common water constituents [125]. This surface-mediated approach enables efficient PFOA degradation even in complex water matrices. Furthermore, ferrous iron activation provides a practical pathway for ambient temperature treatment, achieving 64% PFOA degradation within 4 h under anoxic conditions [126]. These iron-based systems thus address PFAS persistence through combined surface enrichment, localized radical generation, and adaptable activation conditions that collectively enhance degradation efficiency.

The treatment of heavy metal–organic complexes (e.g., metal-EDTA, metal-citrate) presents a unique challenge that requires sequential processing: initial decomplexation to break the stable coordination spheres, followed by immobilization of the liberated metal ions (Figure 7d). Enhanced iron cycling provides an elegant solution to this two-stage problem through sustained generation of reactive species and facilitated electron transfer processes. The decomplexation stage primarily relies on radical-mediated oxidation, where accelerated Fe(II)/Fe(III) cycling in systems such as electrochemical Fe-MOF/persulfate enables continuous production of hydroxyl radicals (•OH) [127]. These strongly oxidizing species selectively attack the coordination bonds in complexes like Cu-EDTA through hydrogen abstraction and ligand oxidation pathways, effectively liberating free Cu^2+^ ions while partially mineralizing the organic ligands. The subsequent immobilization stage then addresses the fate of these freed metal ions through multiple mechanisms: in electrochemical systems, the applied potential drives cathodic reduction and deposition of metal ions, effectively recovering them in elemental form; in parallel, the iron-based catalysts themselves provide adsorption sites and nucleation centers for metal hydroxide precipitation. This integrated approach is further exemplified in Fe(II)-Al layered double hydroxides, where structural Fe–OH sites serve as electron transfer bridges, facilitating both the oxidation of arsenite (As(III)) to arsenate (As(V)) and subsequent adsorption of the transformed arsenic species through surface complexation and structural incorporation [128]. The synergy between enhanced iron cycling and these immobilization pathways ensures not only effective decomplexation but also prevents re-complexation and secondary contamination, offering a comprehensive solution for treating persistent metal–organic complexes that surpasses conventional treatment limitations.

In conclusion, the strategic enhancement of iron cycling is not a universal remedy but a versatile toolkit. The key to precision remediation lies in diagnosing the contaminant’s molecular identity and matching it with a tailored iron cycling strategy—be it to generate powerful radicals, selective non-radical oxidants, or to enable coupled degradation-immobilization processes. This mechanism-based framework provides a rational design principle for developing targeted, efficient, and sustainable water treatment technologies.

### 4.2. Adaptability to Complex Environmental Matrices

The performance of iron-based persulfate systems in real-world applications is critically challenged by complex environmental matrices. Background constituents, such as inorganic anions and natural organic matter (NOM), can scavenge reactive species, compete for active sites, and alter reaction pathways. Strategies that enhance iron cycling demonstrate superior resilience to these interfering species by sustaining catalytic activity through robust electron transfer and surface-mediated radical generation, rather than relying solely on short-lived radicals in bulk solution.

The presence of common anions (e.g., Cl^−^, HCO_3_^−^, H_2_PO_4_^−^) typically quenches free radicals and suppresses degradation efficiency. However, systems with reinforced iron cycling maintain significant activity through enhanced interfacial processes. The magnetic FeS@biochar (MFB-500)/PMS system exhibits strong resilience toward Cl^−^, NO_3_^−^, and SO_4_^2−^ (<20% inhibition in 2,4-D degradation), though performance declines with HCO_3_^−^ and H_2_PO_4_^−^ due to their radical quenching and precipitation effects [123]. The biochar matrix facilitates electron transfer and surface Fe(II) regeneration, thereby reducing dependence on free radicals and enhancing tolerance to anionic interference. Similarly, Sr-doped LaFeO_3_ (LSFO-40) retains over 90% degradation efficiency of Orange I under coexisting Cl^−^ and HCO_3_^−^ across pH 5–9. Doping increases oxygen vacancy concentration, which accelerates Fe(III)/Fe(II) turnover and strengthens PMS adsorption, effectively minimizing competitive anion binding [129].

Natural organic matter such as humic acid (HA) can inhibit degradation by blocking active sites or scavenging reactive species. Nevertheless, systems with enhanced iron cycling demonstrate notable resistance. Magnetic MnFe_2_O_4_ synthesized from spent lithium-ion batteries (MFO-LIBs) achieves 85% bisphenol A degradation even under high HA concentration (20 mg/L) and alkaline conditions (pH 10). The heterostructure provides multiple redox couples (Mn(II)/Mn(III) and Fe(II)/Fe(III)) that maintain catalytic activity with limited influence from HA [24]. In another example, the presence of HA causes only an 8% decrease in bisphenol A degradation efficiency in a CoFe_2_O_4_-LIBs/PMS system. The highly synergistic Co(II)/Co(III) and Fe(II)/Fe(III) cycles compensated for radical consumption by HA, underscoring the role of continuous metal redox cycling in mitigating NOM-induced inhibition [25].

Real wastewater systems introduce additional complexity, including heavy metals and oily constituents, which can further challenge treatment efficiency. In an nZVI-PS system applied to e-waste wastewater, coexisting Cu^2+^ promotes a cooperative redox cycle involving Cu^0^–Cu(II) and Fe(II)/Fe(III), which enhances persulfate activation and increases BDE-47 degradation by 40%. In contrast, Zn^2+^ and Ni^2+^ adsorb onto catalytic surfaces and inhibit reaction efficiency [130]. Similarly, oil-sludge carbon-supported CoFe_2_O_4_ (CoFe_2_O_4_/OSC) achieves 90.8% norfloxacin degradation in an oily water matrix. The dual metal cycling between Co and Fe counteracts scavenging by organic constituents, while metal leaching remains low (<30 μg/L), illustrating the potential of iron-enhanced systems under challenging environmental conditions [131].

In summary, the adaptability of iron-persulfate systems to complex matrices is significantly improved by strategies that enhance iron cycling and electron transfer mechanisms. By reducing reliance on highly scavengeable radicals and promoting surface-mediated or direct electron transfer processes, these advanced systems sustain oxidation capacity and support practical application in realistic water and wastewater treatment scenarios.

### 4.3. Pollutant Degradation Kinetics and Mineralization Efficiency

The enhancement of iron cycling efficiency directly translates to superior performance in pollutant degradation, which is quantitatively analyzed through reaction kinetics and mineralization efficiency. The degradation kinetics in these complex systems are most commonly fitted with a pseudo-first-order model (−ln(C/C_0_) = k_obs_ t), where the observed rate constant (k_obs_) serves as a key metric for the net efficiency of persulfate activation and pollutant oxidation. This model is applicable under the common condition of excess persulfate, where the kinetics are governed by the catalyst-mediated generation of reactive species and their subsequent reaction with the target pollutant. The strategies discussed below enhance k_obs_ by fundamentally accelerating the rate-limiting Fe^3+^/Fe^2+^ cycle. Furthermore, the ultimate remediation goal is mineralization, measured as Total Organic Carbon (TOC) removal, which reflects the complete oxidation of organic pollutants to CO_2_ and water, a process often lagging behind initial pollutant degradation due to the formation of persistent intermediates.

Reductants such as sulfides and polyphenols significantly accelerate the Fe(III)/Fe(II) cycle, providing a continuous supply of Fe^2+^ for persulfate activation. Sulfur-modified ZVI (S-mZVI) activated PMS to degrade sulfamethoxazole (SMX) with an observed rate constant (k_obs_) of 0.1403 min^−1^, representing a 29.4% increase over the unmodified system (Li et al., 2019 [132]). The addition of gallic acid (GA) to a Fe^3+^/PMS system increased the degradation rate constant of BDE47 by 9.4 times (k_obs_ = 0.298 h^−1^), attributable to the GA oxidation intermediates that continuously regenerate Fe^2+^ from Fe^3+^ [133]. The increase in k_obs_ is directly tied to the accelerated Fe^3+^ reduction kinetics provided by these reductants. However, the mineralization efficiency is highly dependent on the nature of the reductant and the resultant reactive species; for instance, reductants that promote a radical-dominated pathway (e.g., SO_4_•^−^ and •OH) typically achieve more complete TOC removal than those favoring selective non-radical pathways.

Bimetallic systems establish complementary redox couples that facilitate electron transfer, markedly improving iron cycling kinetics. In a Cu^0^/Fe_3_O_4_-PMS system, the degradation rate of Rhodamine B increased 36-fold compared to Fe_3_O_4_ alone, due to Cu^0^-mediated reduction of Fe^3+^ and enhanced singlet oxygen production [134]. Pd/Al_2_O_3_ activated PMS for degrading 1,4-dioxane with a metal-loading normalized pseudo-first-order rate constant 16,800 times higher than that of a Cu-Fe bimetallic system, as Pd nanoparticles efficiently recycled surface Fe^2+^ [135]. This dramatic increase in k_obs_ stems from the establishment of a synergistic redox cycle that lowers the activation energy for the critical Fe^2+^ regeneration step. The continuous and concurrent generation of multiple radical species (e.g., SO_4_•^−^, •OH) in these systems is particularly effective for fragmenting aromatic rings and oxidizing intermediate products, thereby bridging the gap between rapid degradation and high mineralization efficiency.

Carbon materials serve as electron mediators that enhance iron cycling through conductive networks and defect-induced catalysis. Graphitized nanodiamond (G-ND) activated PMS for phenolic degradation, achieving a TOC removal >65%, as its sp^2^-carbon surface facilitated direct electron transfer from pollutants to persulfate [136]. Encapsulated nZVI@biochar (BC-Fe^0^) activated PDS to achieve 100% removal of paracetamol within 20 min (k_obs_ = 0.37475 min^−1^), where the carbon shell prevented Fe^0^ passivation and promoted non-radical pathways [137].

External energy fields such as weak magnetic fields (WMFs) and light irradiation overcome diffusion and kinetic barriers in iron-activated systems. The application of a WMF to a Fe^0^/PMS system increased the k_obs_ for SMX degradation by 2–3 times, owing to magnetically enhanced convection that accelerated Fe^0^ corrosion and Fe^2+^ release [138]. While effective, these approaches often require additional infrastructure and energy input, and their effectiveness may be limited for non-magnetic or homogeneous catalysts.

In summary, the enhancement of iron cycling unequivocally accelerates pollutant degradation kinetics (k_obs_) by overcoming the electron-transfer bottleneck. More profoundly, the pathway to efficient mineralization is dictated by the nature of the dominant oxidizing species sustained by these strategies. Systems that foster a high and continuous flux of strong radicals, particularly SO_4_•^−^ and •OH, generally achieve superior TOC removal. This is because these non-selective radicals are uniquely capable of unselectively attacking and fragmenting the diverse and recalcitrant intermediate compounds that accumulate during oxidation. In contrast, while highly selective, non-radical pathways (e.g., ^1^O_2_, direct electron transfer) often exhibit lower mineralization potentials. Therefore, the rational selection of an iron-cycle enhancement strategy must balance the goal of rapid initial degradation (high k_obs_) with the requirement for deep oxidation (high TOC removal), by deliberately steering the reaction pathway towards radical generation where complete mineralization is the priority.

### 4.4. Catalyst Stability and Recyclability in Iron-Enhanced Persulfate Systems

The transition from laboratory research to practical application of iron-based persulfate systems critically depends on catalyst stability and recyclability. Enhanced iron cycling strategies directly address the fundamental challenges of active site preservation and structural integrity maintenance over extended operational periods [139,140]. These advancements primarily operate through the introduction of auxiliary redox couples, strategic physical confinement of active components, and the innovative design of self-regenerative interfaces—all contributing to sustained catalytic performance.

Multivalent elemental doping significantly improves stability by establishing complementary redox systems that reduce dependence on iron dissolution. In CuFe_2_O_4_-CoFe_2_O_4_ composites, synergistic interactions between Co^2+^/Co^3+^ and Cu^+^/Cu^2+^ cycles promote efficient Fe^2+^/Fe^3+^ cycling while markedly reducing metal leaching, maintaining 94.3% degradation efficiency after five operational cycles [139]. The stabilization of coordinatively unsaturated iron sites in mixed-valence systems similarly demonstrates how alternative redox pathways can alleviate irreversible iron loss, though potential environmental implications of secondary metal leaching require careful consideration [140].

Structural confinement strategies effectively immobilize active iron species through tailored support interactions and engineered interfaces. Encapsulation of iron oxides within carbonaceous matrices suppresses metal leaching via strong electronic interactions, with some systems demonstrating iron loss rates below 0.125 wt% while retaining over 87% catalytic activity after multiple cycles [141]. Biochar-supported systems utilize both dispersive effects and surface functional groups to stabilize iron species, achieving minimal leaching through combined physical and chemical stabilization mechanisms [142]. The critical importance of interfacial bonding strength is increasingly recognized, as weaker interactions may lead to structural degradation under challenging operational conditions [143].

Advanced self-regenerative systems represent a paradigm shift in catalyst design, where cocatalysts enable continuous iron cycling through interfacial electron transfer. Metal sulfides such as MoS_2_ function as effective electron mediators, reducing Fe^3+^ to Fe^2+^ while undergoing controlled oxidation themselves, decreasing iron leaching rates to 0.33% of initial levels while maintaining >80% efficiency over six cycles [144]. WS_2_ as a cocatalyst enhances electron transfer via surface sulfur vacancies, improving iron cycle efficiency tenfold [145]. The long-term efficacy of these systems depends critically on the chemical stability of the cocatalyst components, as structural transformations under operational conditions may gradually diminish their electron transfer capability [146].

The regeneration of spent catalysts is a key factor for long-term operation. Common regeneration methods include simple water washing to remove surface-adsorbed species, thermal treatment (e.g., annealing at 300–500 °C in inert atmosphere) to restore surface properties and remove carbonaceous deposits, and chemical washing with solvents or mild oxidants. However, catalysts invariably experience varying degrees of activity loss upon regeneration, primarily due to irreversible structural changes, metal leaching accumulation, and the inability to fully restore the original active site coordination. The reported number of effective regeneration cycles in the literature is typically in the range of 3 to 6 cycles while maintaining >80% of the initial activity, as exemplified by systems like Fe/Mo@C [48] and FeCo@NBC [51]. More robust designs, such as those employing strong structural confinement [141], demonstrate the potential for extended stability over even more cycles, though a gradual decline in performance is generally observed due to the aforementioned irreversible changes.

Despite these significant advancements, the field requires more rigorous validation under environmentally relevant conditions. Most current studies report only short-term cycling performance, leaving open questions about long-term stability in complex aqueous matrices containing high salinity or organic loadings [129]. Furthermore, standardization of leaching assessment protocols is urgently needed, as current methodologies vary widely in their detection limits and speciation capabilities. Future research should prioritize in situ characterization of iron valence states and structural evolution during extended operation, providing deeper insights into deactivation mechanisms and regeneration possibilities.

In conclusion, enhancing iron cycling through multivalent doping, structural confinement, and regenerative interfaces provides a multifaceted strategy for improving catalyst stability and recyclability. The continued advancement of these systems toward practical implementation will require combined efforts in materials design, operational optimization, and standardized stability assessment under realistic environmental conditions.

### 4.5. Environmental Risk and Cost-Effectiveness Analysis of Iron-Cycle Enhanced Systems

While iron-cycle enhancement strategies significantly improve the catalytic performance of persulfate-based advanced oxidation processes, their practical implementation requires careful evaluation of environmental risks and economic feasibility. A comprehensive analysis reveals several critical challenges that must be addressed before large-scale application.

Environmental risk considerations primarily focus on metal leaching and transformation byproduct formation. Although carrier immobilization strategies such as biochar-supported Fe_3_O_4_ can reduce iron leaching to below 0.125 wt% after multiple cycles, bimetallic systems introduce additional concerns regarding secondary metal leaching (e.g., Co, Cu) that may cause secondary pollution [139]. More importantly, the alteration of degradation pathways due to enhanced iron cycling may generate more toxic intermediates than the parent compounds. Radical-dominated reactions tend to produce hazardous chlorinated quinones and nitro-byproducts during antibiotic degradation, while non-radical pathways, though potentially reducing halogenated byproducts, rely on the stability of high-valent metal species whose fluctuation may increase treatment uncertainty [27,35,36]. Furthermore, iron precipitates such as Fe(OH)_3_ flocs can adsorb toxic transformation products, forming “iron-toxin” co-precipitates that accumulate in sediments and pose potential ecological risks [39].

Economic feasibility analysis indicates that cost considerations extend beyond simple catalyst synthesis to encompass operational expenses and long-term sustainability. High-performance catalysts including Fe/Mo-CNs and Sr-doped LaFeO_3_ require complex preparation procedures that increase material costs [13,129]. Electrochemically assisted strategies, while effective in accelerating iron cycling, consume substantial energy (up to 2.51 kWh/m^3^), significantly exceeding the costs of conventional Fenton systems [147]. The continuous addition of chemical reductants (e.g., hydroxylamine, cysteine) further increases operational costs and raises questions about long-term economic viability [148,149].

Promising strategies for enhanced sustainability emerge through waste resource utilization and renewable energy integration. Catalysts derived from waste lithium batteries or sludge-derived biochar not only reduce raw material costs by 30–50% but also enable value-added use of solid waste [25,150]. The incorporation of crystalline boron, despite increasing reaction rates by 65-fold, raises concerns about potential water eutrophication due to boron residue, highlighting the need for comprehensive environmental risk assessment even for effective enhancers [151]. Similarly, while MoO_2_ modification effectively promotes Fe^2+^/Fe^3+^ cycling, the long-term ecotoxicity of molybdenum remains insufficiently studied, requiring further investigation into its environmental fate and impact [13].

In summary, while iron-cycle enhancement strategies show remarkable improvements in catalytic performance, their practical implementation necessitates a balanced consideration of environmental risks and economic factors. Future advancements in this field should focus on developing comprehensive assessment protocols that simultaneously address treatment efficiency, economic viability, and environmental safety to ensure the sustainable development of iron-persulfate systems for water treatment applications.

### 4.6. Comparative Analysis with Conventional AOPs

A systematic comparison reveals both advantages and limitations of iron redox cycling-based persulfate processes relative to established AOPs. In terms of degradation efficiency, these systems generate multiple reactive species including sulfate radicals (SO_4_•^−^, E^0^ = 2.5–3.1 V), hydroxyl radicals (•OH), and high-valent iron species, enabling effective degradation of recalcitrant compounds that resist conventional •OH-based oxidation [1,10,15]. The SO_4_•^−^ demonstrates particular effectiveness against electron-rich organic contaminants due to its selective oxidation mechanism and longer half-life compared to •OH [15,31].

Operationally, iron-persulfate systems function effectively across pH 3–9, significantly broader than classical Fenton’s narrow optimal range (pH 2.5–3.5) where iron precipitation becomes problematic [4,6,12]. While homogeneous Fe^2+^/persulfate systems operate with minimal energy input, the enhanced systems employing thermal activation (60–80 °C) or electrochemical assistance incur substantial energy costs that may approach those of UV/H_2_O_2_ processes [100,147].

Environmental considerations present distinct challenges. Unlike Fenton processes that generate substantial iron sludge (often 10–30% of treated water volume), properly designed heterogeneous systems significantly reduce solid waste [4,6]. However, sulfate accumulation from persulfate decomposition (typically 200–500 mg/L SO_4_^2−^ per 1 g/L persulfate dose) represents a unique concern absent in H_2_O_2_-based systems [15]. Furthermore, advanced bimetallic catalysts, while enhancing iron cycling efficiency, introduce risks of secondary metal leaching (e.g., Co, Cu) that require careful management [129,139].

From a practical perspective, these systems show particular promise for in situ remediation scenarios where persulfate’s subsurface stability (half-life of weeks to months) provides significant advantages over the rapid decomposition of H_2_O_2_ (half-life of hours) [1,10]. They also demonstrate superior performance for treating halogenated compounds and certain pharmaceuticals where SO_4_•^−^ exhibits higher oxidation efficiency than •OH [15,31]. While full-scale implementation currently lags behind established technologies like ozonation and conventional Fenton, ongoing advances in catalyst design and reactor engineering are rapidly closing this gap.

This analysis confirms that iron redox cycling-based persulfate processes occupy a valuable niche in the AOP spectrum, particularly suited for challenging remediation scenarios where their multi-oxidant capacity and operational flexibility provide distinct advantages, provided that sulfate accumulation and potential metal leaching are properly managed.

## 5. Challenges and Future Perspectives

The advancement of iron redox cycling for persulfate activation presents a complex interplay between promising laboratory demonstrations and the multifaceted realities of environmental application. This section critically examines the current challenges rooted in mechanistic, material, and system-level constraints, and reframes future research needs as concrete opportunities to bridge the gap between fundamental insight and sustainable implementation.

### 5.1. Current Research Challenges

While numerous studies have demonstrated enhanced performance through various strategies, the precise mechanisms governing interfacial electron transfer and radical/non-radical pathway selection remain inadequately resolved. This complexity, arising from the generation of multiple reactive species (SO_4_•^−^, •OH, ^1^O_2_, Fe(IV)=O), can offer a broader reactivity profile against recalcitrant pollutants compared to the single-radical •OH-dominated chemistry of conventional Fenton systems [4], but it concurrently complicates mechanistic elucidation and control. The complexity of multiphase systems, where homogeneous and heterogeneous processes coexist, creates significant challenges in decoupling individual contribution mechanisms. The lack of standardized protocols for identifying reactive oxygen species further complicates cross-study comparisons and mechanistic validation.

The current technological readiness level of most advanced iron-persulfate systems remains low, with complex catalysts like single-atom Fe-N-C sites and core–shell heterostructures typically demonstrating only single-digit operational cycles in laboratory studies [56,60,74]. This represents a significant gap from the >50 cycles required for practical water treatment applications. Many high-performance catalysts rely on precious or toxic metals (e.g., Co, Mo, Pd) and energy-intensive synthesis methods, raising concerns about environmental footprint and scalability. This reality contrasts sharply with the ideal of sustainable environmental remediation. Economic viability analysis reveals substantial challenges, with electrochemical assistance consuming significantly more energy than conventional treatment [147], while continuous reductant addition increases operational costs [148]. The synthesis of complex nanostructures often involves multiple steps, organic solvents, and high-temperature treatments that are difficult to scale economically. Moreover, the long-term stability of these advanced materials under realistic environmental conditions remains largely unverified, with most studies reporting fewer than ten operational cycles under idealized laboratory conditions.

A pronounced performance gap exists between laboratory and field conditions, where degradation efficiency typically decreases significantly in real wastewater due to radical scavenging by background constituents [123,129]. Similarly, iron leaching rates increase substantially when transitioning from ultrapure water to real wastewater matrices [139,142]. The transition from batch-scale experiments to continuous-flow systems presents substantial engineering hurdles. Mass transfer limitations, catalyst recovery and reuse, and energy input requirements become critical factors at larger scales. Additionally, the presence of complex matrices in real wastewater—including suspended solids, natural organic matter, and competing ions—can significantly diminish system performance through catalyst fouling, radical scavenging, and active site blockage.

Current research predominantly focuses on performance metrics while paying insufficient attention to potential secondary pollution risks. The leaching of metal ions (both iron and dopants), formation of toxic transformation products, and long-term ecotoxicity of catalyst components require comprehensive evaluation. As discussed in Section 4.5, secondary metal leaching from bimetallic systems and the formation of toxic byproducts represent significant concerns that are rarely addressed in current studies [35,139]. Life cycle assessment studies are notably absent from the literature, leaving a critical gap in understanding the overall environmental footprint of these enhanced systems.

### 5.2. Future Research Directions

To propel the iron-persulfate systems from fundamental research toward practical implementation, future endeavors should pivot to addressing the core mechanistic and application bottlenecks through cutting-edge operando techniques, sustainable material design, and smart process integration. Especially, future research should prioritize the application of in situ/operando techniques (e.g., in situ Raman, X-ray absorption spectroscopy, Mössbauer spectroscopy) to dynamically monitor iron valence states, surface transformations, and reaction intermediates under operational conditions. This represents a crucial opportunity to resolve the mechanistic ambiguities highlighted in Section 3.1, Section 3.2, and Section 3.3, particularly for understanding interface electron transfer in heterostructures [53,57] and vacancy-mediated activation in defect-engineered catalysts [22,62]. Coupling these experimental approaches with computational methods (DFT, machine learning) can establish predictive structure-activity relationships and guide the rational design of next-generation catalysts with optimized iron cycling capabilities.

The current reliance on scarce or toxic metals presents a compelling opportunity to pioneer a new paradigm of catalysts designed for circularity and minimal environmental impact. Research should focus on developing earth-abundant, environmentally benign catalysts using green synthesis approaches. This includes transforming waste streams into valuable catalytic materials, thereby addressing both pollution control and waste management challenges. Waste-derived materials (e.g., sludge biochar [150], spent battery materials [24,25]) offer promising avenues for reducing costs and environmental impacts while maintaining performance. Design strategies should emphasize structural robustness, easy separation, and regeneration capabilities to enhance practical applicability.

To realize this sustainable catalytic paradigm, several advanced material design strategies are emerging at the forefront. Beyond merely replacing critical metals, these approaches aim to maximize the activity and longevity of earth-abundant elements. For instance, engineering single-atom sites (e.g., Fe-N_4_) within conductive matrices can achieve near-optimal atom utilization while minimizing leaching, directly supporting the goal of minimal environmental impact [20,59]. Similarly, constructing confined nanoenvironments (e.g., within nanocages or MOFs) can protect active iron species from aggregation and poisoning, thereby enhancing the structural robustness and lifetime emphasized above [74,141]. Furthermore, the development of self-driven or energy-adaptive systems represents a pivotal step toward reducing operational energy costs, aligning economic and environmental sustainability [147,148]. Collectively, these advanced architectures provide the technological toolkit to build the next generation of high-performance yet sustainable iron-based persulfate catalysts.

The limitation of standalone AOPs in complex matrices can be transformed into an opportunity by developing intelligent, integrated process trains. Future work should explore the integration of iron-enhanced persulfate systems with complementary technologies (e.g., membrane filtration, biological treatment) to create synergistic treatment trains. Research on reactor design, process control, and energy optimization for continuous-flow operation is essential for technological maturation. Additionally, developing stimuli-responsive or adaptive systems that can maintain efficiency under fluctuating water quality conditions represents an important frontier.

The gap between laboratory benchmarks and field application underscores the critical need for standardized, environmentally focused evaluation protocols. Establishing standardized testing protocols for evaluating catalyst stability, metal leaching, and transformation product formation is crucial for meaningful cross-study comparisons. Research must expand beyond efficiency metrics to include thorough ecotoxicity assessments and life cycle analyses to ensure overall environmental benefits. Long-term field studies in real-world environments are necessary to validate laboratory findings and assess practical feasibility.

The significant energy footprint of external field-assisted strategies represents a major opportunity to align advanced oxidation processes with global carbon neutrality goals. Given the energy requirements of some enhancement strategies (e.g., electrochemical, UV assistance), future research should focus on coupling these systems with renewable energy sources. Solar-driven activation, particularly through photothermal and photovoltaic integration, represents a promising pathway toward carbon-neutral water treatment operations.

## 6. Conclusions

This review has systematically established that the kinetics of iron cycling constitute the central bottleneck in iron-persulfate oxidation systems. Through critical examination of three primary enhancement strategies—material design, system optimization, and external field assistance—we have established a comprehensive framework that links atomic-scale modifications to macroscopic performance improvements. Our analysis reveals that these strategies effectively enhance iron cycling through three convergent mechanisms: (1) facilitating electron transfer via conductive networks, redox mediators, or external energy fields; (2) suppressing iron precipitation through structural confinement, surface modification, and complexation; and (3) precisely regulating reaction pathway selection between radical and non-radical routes through electronic structure modulation. The integration of advanced characterization techniques and theoretical modeling has been crucial in elucidating these fundamental mechanisms at the molecular level. The remarkable progress in iron cycling enhancement has yielded substantial practical benefits, including unprecedented degradation efficiency for diverse contaminant classes, expanded operational pH windows, and significantly improved oxidant utilization rates. Specific material designs have demonstrated exceptional performance, with some heterostructured catalysts achieving up to 36-fold increases in degradation rates while maintaining stability over multiple cycles. However, our analysis reveals that future research must address several critical challenges to enable practical implementation. The development of standardized assessment protocols for evaluating long-term stability, environmental impacts, and economic feasibility is essential for meaningful technology advancement. Particular attention should be directed toward understanding the fate and transformation of catalyst components and reaction intermediates in complex environmental matrices, as well as managing sulfate accumulation in treated water. Looking forward, the field must embrace a holistic approach that balances performance optimization with sustainability considerations. The integration of renewable energy sources, waste-derived materials, and intelligent system designs represents a promising pathway toward sustainable water treatment applications. Future research should prioritize interdisciplinary collaborations that bridge materials science, environmental engineering, and computational methods to accelerate the development of next-generation iron-based persulfate systems. Ultimately, the rational design of iron-enhanced persulfate systems based on fundamental mechanistic understanding will enable transformative advances in water purification technologies. By addressing both scientific and practical challenges through integrated research efforts, we can realize the full potential of these systems for addressing emerging contaminant removal while maintaining environmental and economic sustainability.

## Figures and Tables

**Figure 1 nanomaterials-15-01712-f001:**
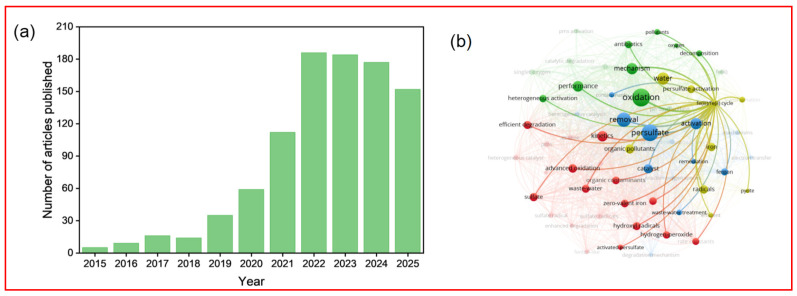
(**a**) Annual number of publications on iron-catalyzed persulfate activation for environmental remediation, indicating consistent growth and increasing research interest since 2015. (**b**) Keyword co-occurrence network analysis visualizing major research themes and conceptual relationships in iron-enhanced persulfate activation systems.

**Figure 2 nanomaterials-15-01712-f002:**
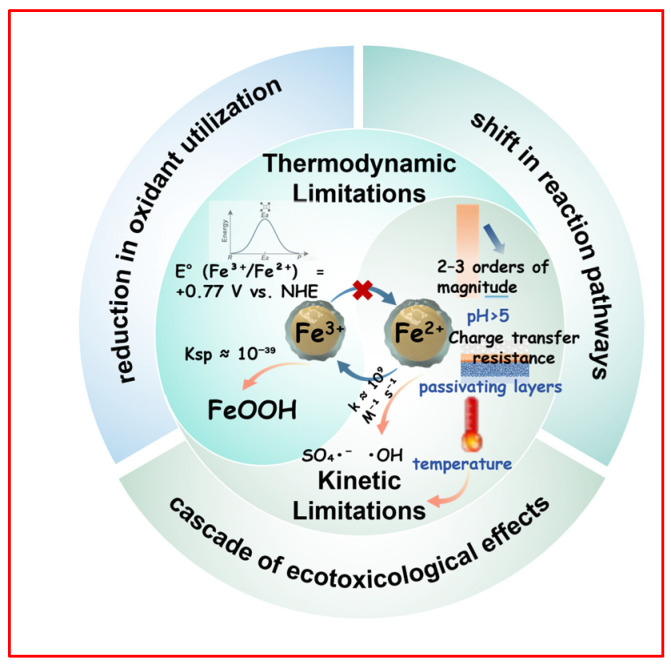
The Root Causes and Impacts of Low Iron Cycling Efficiency.

**Figure 3 nanomaterials-15-01712-f003:**
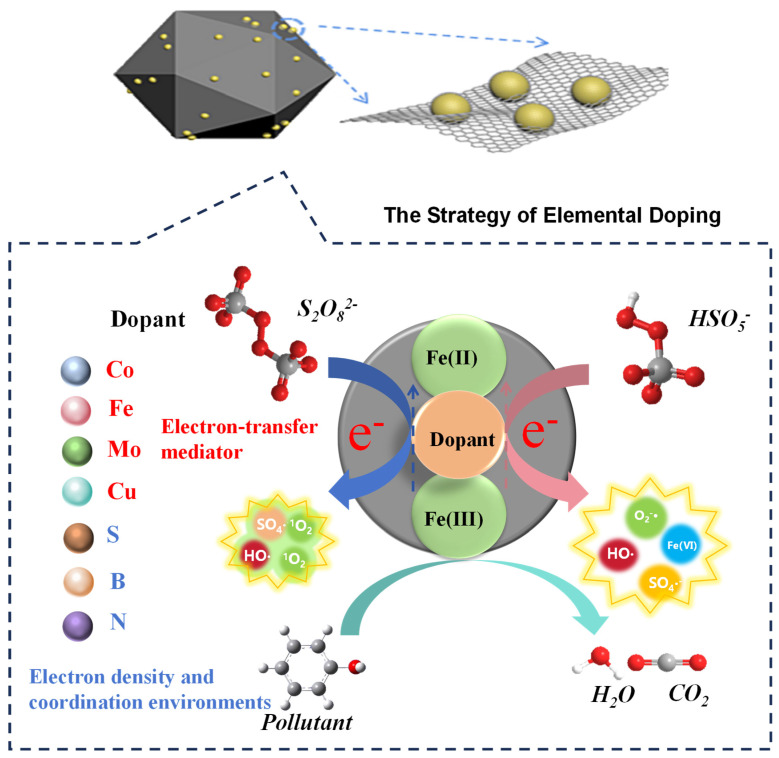
Electronic structure modulation via elemental doping to produce ROS.

**Figure 4 nanomaterials-15-01712-f004:**
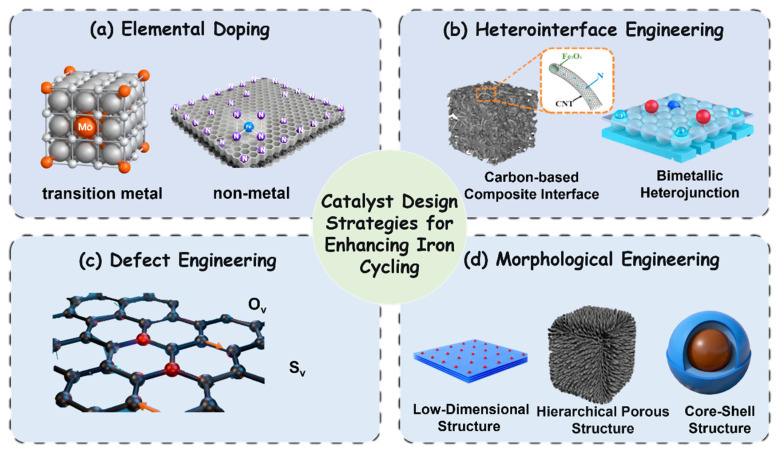
Catalyst Design and Modification Strategies for Enhanced Iron Cycling. (**a**) Elemental Doping. (**b**) Heterointerface Engineering. (**c**) Defect Engineering. (**d**) Morphological Engineering.

**Figure 5 nanomaterials-15-01712-f005:**
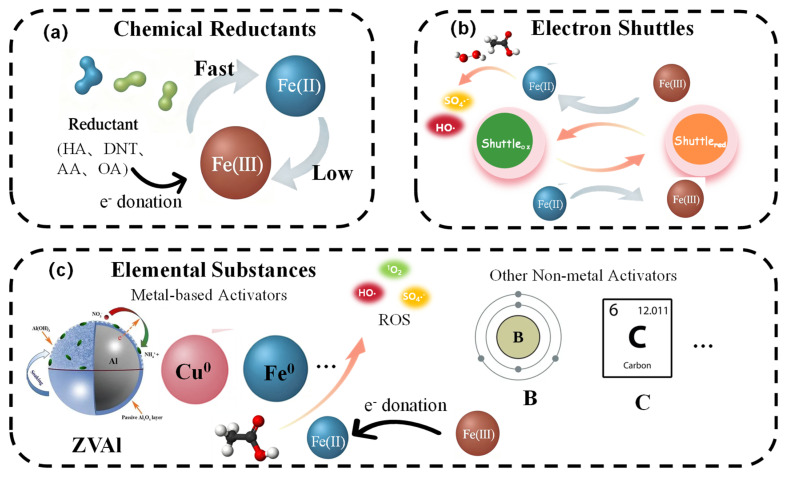
Optimization Strategies via Reductive Enhancement. (**a**) Chemical Reductant. (**b**) Electron Shuttles. (**c**) Elemental Substances.

**Figure 6 nanomaterials-15-01712-f006:**
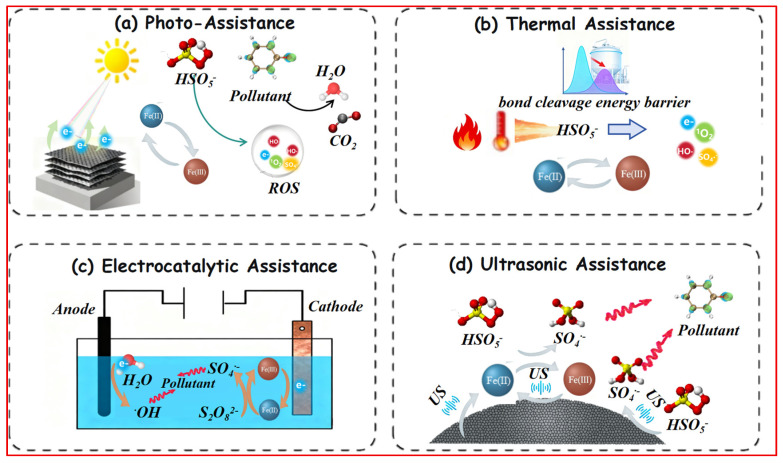
External Field Assistance Strategies. (**a**) The photocatalytic mechanism. (**b**) Thermal Activation. (**c**) Electrocatalytic Assistance. (**d**) Ultrasound assistance.

**Figure 7 nanomaterials-15-01712-f007:**
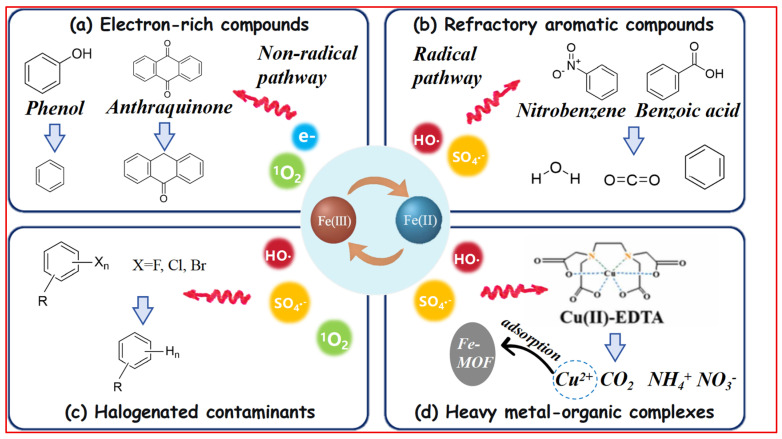
Mechanism-Tailored Strategies for Different Contaminant Classes. (**a**) The Electron-rich compound degradation. (**b**) The Refractory aromatic compounds. (**c**) The halogenated contaminants. (**d**) The Heavy metal–organic complexes.

**Table 1 nanomaterials-15-01712-t001:** Fundamental Limitations Governing Inefficient Iron Cycling in Persulfate Activation Systems.

Intrinsic Nature	Specific Manifestation & Mechanism	Impact on Catalytic Performance & System Efficiency	References
Kinetic Limitations	The intrinsic electron transfer rate for the reduction of Fe^3+^ (especially hydrolyzed species, e.g., Fe(OH)^2+^) is prohibitively slow	Becomes the unequivocal rate-limiting step, causing a precipitous drop in radical flux and extending treatment half-lives by 1–2 orders of magnitude.	[12,15]
Kinetic Limitations	Fe^2+^ acts as a potent scavenger for SO_4_•^−^/•OH radicals (k ≈ 10^9^ M^−1^s^−1^), competing with target pollutants.	Leads to significant oxidant waste (utilization efficiency drops by 70–80%) and rapidly depletes the dissolved Fe^2+^ pool.	[31,44]
Thermodynamic Limitations	Hydrolysis of Fe^3+^ forms insoluble (oxyhydr)oxides (e.g., FeOOH), representing the thermodynamically stable state in aqueous media, especially at neutral-alkaline pH.	Depletes bioavailable iron, terminating the homogeneous cycle. Traps iron in a solid phase, making reduction unfavorable.	[13,15]
Thermodynamic Limitations	Formation of insoluble Fe^3+^ salts (e.g., FePO_4_) sequesters Fe^3+^ via strong coordination bonds (Ksp ≈ 10^−22^), thermodynamically stabilizing the Fe^3+^ state and lowering the effective Fe^3+^/Fe^2+^ redox potential from +0.77 V to ~+ 0.36 V.	Thermodynamically suppresses the driving force for Fe^2+^ regeneration, exacerbating the kinetic bottleneck.	[39]
Interfacial & System-Level Constraints	The precipitation and deposition of iron oxides onto heterogeneous catalysts form a physical barrier, drastically increasing charge transfer resistance.	Blocks active sites and impedes electron transfer from the bulk to the surface, leading to rapid catalyst deactivation.	[41,42]
Interfacial & System-Level Constraints	Cycle stagnation shifts oxidation pathway, which exhibit different selectivity.	May lead to incomplete degradation and the generation of more toxic transformation products.	[27,35,36]

## Data Availability

No new data were created or analyzed in this study. Data sharing is not applicable to this article.

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
