# Peer review of "Iron Redox Cycling in Persulfate Activation: Strategic Enhancements, Mechanistic Insights, and Environmental Applications—A Review"

_nanomaterials, 2025, doi:10.3390/nano15221712_

Round 1

Reviewer 1 Report

Comments and Suggestions for Authors

The article titled "Iron Redox Cycling in Persulfate Activation: Strategic Enhancements, Mechanistic Insights, and Environmental Applications" has potential for acceptance, after some minor corrections:

1) The keywords are too numerous. Please review these.

2) Figure 1b is completely ineligible. Please modify or delete it.

3) Lines 169 to 176 are difficult to understand and require revision.

4) Take inspiration from Figure 4; it's very good. It's very easy to understand.

Otherwise, the article is very well written.

Reviewer 2 Report

Comments and Suggestions for Authors

1) The content of this review is information-rich and covers a wide range of topics; however, it does not effectively reflect the concept of ’’ Mechanistic Insights’’; as emphasized in the title. Most sections are presented in a descriptive or listing style, lacking the critical analysis and synthesis that are expected in a review article.

2) Every figure involves one article and thus lacks of generality, which is also obvious in the corresponding text of the article. The description of the data for one article is too specific. The authors are suggested to make a summary of the previous reports and select several representatives for presentation.

3) More figures are suggested to be added in the revised manuscript with respect to the environmental applications. The figure quality needs to be improved.

4) The Reality, challenges and opportunities of Iron Redox Cycling in Persulfate Activation should be discussed in the revised manuscipt, as it is very important. It should be added in place of future prospects.

Reviewer 3 Report

Comments and Suggestions for Authors

The paper broadly covers the Iron Redox Cycling-persulfate-based processes for degradation of the organics. The authors should emphasize the novelty and difference in the approach compared to similar reviews. The paper should be more coherent; some features should be better and more deeply explained and discussed. The kinetic part should provide kinetic models. The literature used is relevant, but some more is probably needed. The legend on some graphs is too small, unclear, and hardly visible. The language, writing style, and grammar are satisfying.

The specific comments to the authors to improve the manuscript quality are as follows:

  1. This looks like a review paper. If the paper was submitted like that, the word “Review” should be in the title.
  2. Introduction, line 72: Sentence “/urge of research interest in persulfate activation using iron-based catalysts.” has a problematic beginning. Please check.
  3. Introduction: The authors should clearly emphasize the novelty and difference of this paper, especially compared to similar reviews.
  4. Figure 1: Letters in Figure 1b are small and unclear. Please make them more visible.
  5. The paper should include the analysis of the possible practical applications. Feasibility, capacity, cost, challenges, and drawbacks.
  6. Figure 2: Some of the letters and small graph on the left side of the image are too small and barely visible. Please make them more visible and clearer.
  7. Section 2.2, lines 170-176: The claims must be backed by the references at this place.
  8. Table 1: “Thermodynamic” instead of “Thermo dynamic”. Correct everywhere it appears in the text.
  9. Section 3.1.1: The incorporation of transition metals, wherein the dopant’s valence cycle acts as the electron-transfer mediator, should be essentially theoretically presented and explained before passing to the examples.
  10. Section 3.1.1: The last paragraph must be backed by the proper literature.
  11. Figure 3 b and d: The legend is too small; please make it clearer and more visible. The same for Zn2+ structure (blue) in Figure 3e (it should be drawn in some color more contrasted to the background).
  12. Sections 3.1.1 and 3.1.2 should contain a general rule (theoretical or empirical, if any) of how to choose the proper dopant to obtain a desired electronic structure/properties of the material. What precursors for doping could be used, and based on what should they be chosen? Theory, calculations, experience…?
  13. Section 3.1: The basic synthetic routes for material modifications mentioned in this Section should be numbered and briefly presented, with the advantages and drawbacks of each. For the nanostructures, the most used synthetic routes, as well as the ones most commonly used for this particular purpose (synthesis and modification of nano-structured iron-based catalysts).
  14. Sections 3.1 and 3.2 are similar and overlap in many parts. The authors should edit them by focusing on the distinct aspects and emphasizing the key points of both sections as much as possible.
  15. Figure 5b, right part: the graphs (and the corresponding letters on them) are too small. Please try to redesign the image to make them bigger.
  16. Section 3.3: Only the drawbacks of electrocatalytic activation were mentioned in the text. Please mention the drawbacks of each external activation method mentioned in this Section (relying on Table 4, for example).
  17. Figure 6 a) and c): The letters should be bigger if possible.
  18. Section 4.1: The statements:” Electron-rich compounds including phenolics and dyes undergo preferential degradation through non-radical pathways” (line 695) and: “The sustained generation of SO4•. and •OH proves essential for aromatic ring cleavage and antibiotic inactivation” (line 703) are contradictory. Please consult the literature carefully and address the issue.
  19. Section 4.1, line 717: “For persistent PFAS, conventional radicals prove inadequate, necessitating alternative mechanisms including surface-confined high-valent iron species or enhanced electron transfer pathways targeting fluorinated compound stability”. Please, explain in more detail and more concretely.
  20. Section 4.1: The 5th paragraph (Heavy metal-organic complexes) looks scarce, confusing, and unfinished. Please elaborate it more carefully and in more detail.
  21. Figure 7: The letters in Figure 7d are too small. Please make them more visible or divide the Figure into two figures.
  22. Section 4.3 contains mostly examples, without a deep, comprehensive, and coherent analysis of the problematic. The section should be focused on these reactions’ kinetics itself, not just the ways to improve their rates. What kinetic models do these reactions follow? Also, there is very little about mineralization efficiency. What is it in general, and how can it be improved? What radicals (or other species) cause higher mineralization in general? This all must be addressed.
  23. Section 4.4 should briefly report the regeneration methods of the various iron cycle-based catalysts. Do they lose some of their activity by regeneration? How many times can they be regenerated?
  24. Section 5.2: If some literature was used for this Section, it should be added.
  25. Where do the Iron Redox Cycling-based processes for water treatment stand compared to other common AOPs (efficiency, free radicals that participate in degradation, energy cost, environmental concern, practical applicability)?

Reviewer 4 Report

Comments and Suggestions for Authors

The authors of the manuscript with title "Iron Redox Cycling in Persulfate Activation: Strategic Enhancement, Mechanistic Insights, and Environmental Applications" reports interesting research results. However, there are several drawbacks in the manuscript. The manuscript can be revised taking into consideration comments of the reviewer.

  1. The abstraction section should be modified with the inclusion of the method of activation applied for bperoxymonosulfate and peroxydisulfate.
  2. Introduction should be modified with inclusion of a paragraph about the latest research results published in the open literature regading Iron Redox Cycling in Persulfate Activation for application in Environmental problem solutions.
  3. Figure 2. Root causes and impacts of low Iron cycling efficiency. Effect of temperature, duration of iron cycling should be included in the Figure 2.
  4. Table 1. Fundamental Limitations Governing Inefficient Iron Cycling in Persulfate Activation Systems. Authors are advised to explain the phenomena for the formation of insouble Fe3+ salts (FePO4) 
  5. Figure 3. Electronic structure modulation via Elemental Doping. a) Schematic illustration of the fabrication of Fe/MoC. c) Schematic diagram of the activation mechanism in the Fe/S-NC/PMS system. It suggested authors describe in detail effect of temperature and duration of activation for persufate activation.
  6. Figure 4. Catalyst design and modification strategies for enhanced Iron Cycling. Authors are advised to include more information and latest published research regarding Catalyst design and morphological engineering 
  7. Figure 5. System Optimization strategies via Reductive Enhancement. It is suggested that authors include different types of chemical reductants applied for persulfate ativation
  8. Figure 6. External Field Assistance Strategies. b) Acceleration of Persulfate Activation by ML-101 (Fe) with Vaccum Thermal Activation. Authors should describe in detail the Advantages and Disadvantages of Vaccum Thermal Activation
  9. Figure 7. Mechanism Tailored Strategies for Different Classes . It is suggested that authors include a paragraph with new references regarding Halogenated contaminants and Heavy metal-organics
  10. Challenges anf Future Persepectives. It is suggested that authors include one paragraph with application of new technologies for in persulfate activation
  11. Conclusion section should be modified with important results of the manuscript
  12. Quality of Figures and Tables should be improved
  13. Layout of the manuscript should be improved according to the recommendation of Journal Nanomaterials 

Round 2

Reviewer 3 Report

Comments and Suggestions for Authors

The authors addressed all the comments and suggestions, improving the quality of the paper enough to be published in Nanomaterials

Reviewer 4 Report

Comments and Suggestions for Authors

The authors of the revised version of the manuscript "Iron redox cycling in persulfate activation : strategic enhancements, mechanistic insights, and environmental applications" have taken into consideration comments of the reviewer.

The manuscript is recommended for publication in Catalysts.